# Design and Control of a DC Collection System for Modular-Based Direct Electromechanical Drive Turbines in High Voltage Direct Current Transmission

**Muhammad Ramadan Bin Mohamad Saifuddin** *,†, , **Thaiyal Naayagi Ramasamy** † and **Wesley Poh Qi Tong** †

School of Electrical and Electronic Engineering, Newcastle University, Singapore 567739, Singapore; naayagi.ramasamy@ncl.ac.uk (T.N.R.); w.q.t.poh2@newcastle.ac.uk (W.P.Q.T.)

* Correspondence: m.r.bin-mohamad-saifuddin@ncl.ac.uk
† These authors contributed equally to this work.

**Abstract:** In response to an increasing demand for offshore turbine-based technology installations, this paper proposes to design a DC collection system for multi-connected direct drive turbines. Using tidal stream farm as the testbed model, inverter design and turbine control features were modelled in compliance with high voltage ride-through capabilities that operate in isochronous mode suggested by IEEE1547-2018. The aim of the paper is twofold. Firstly, operation analyses in engaging a single-stage impedance source inverter as an AC-link busbar aggregator to pilot a parallel-connected electromechanical drive system. It uses a closed-loop voltage controller to secure voltage-active power (Volt/Watt) dynamics in correspondence with turbine's arbitrary output voltage level. It also aspires to truncate active rectification stages at generation-side as opposed to a traditional back-to-back converter. Secondly, a proposition for a torque-controlled blade pitching system is modelled to render a close to maximum power point tracking using blade elevation and mechanical speed manipulations. The reserve active power generation aids with compensating an over-voltage crisis as a substitute for typical reactive power absorption. The proposed Testbed system was modelled in PSCAD, adopting industrial related specifications and real-time ocean current profiles for HVDC transmission operations. Analytical results have shown a positive performance index and transient responses at respective tidal steam turbine clusters that observe fault ride-through criterion despite assertive operating conditions.

**Keywords:** energy conversion; solid state circuits; variable speed drives; fault tolerant control; HVDC transmission

## 1. Introduction

A revolution in harnessing ocean current energies into electricity needs exceptional advancement to expedite its maturity towards the future's energy mix. In September 2016, Atlantis Resources have successfully commissioned its first micro-scaled tidal stream farm in Pentland Firth [1] as a testbed system. It was projected to generate a total power of 6 MW from the four installed 1.5 MW tidal stream turbine (TST) systems. Distinct from such initiations, investors in power utilities around the world have consent that tidal power time has finally arrived [2,3]. As tidal current profiles are classified as a predictive source of energy that oscillates in a habitual pattern at a specific time and day, research and statistical analyses have prophesied that TST technology will take its precedence in generating 15% of UK's renewable electricity

supply chain [4,5]. Experts suggest that augmentation in TST technology must be prioritized to transcend supremacy against other offshore energy harvesting avenues [6] for mainland electrifications. Therefore, TST developers must radically rethink new ecotechnological and dependable operational solutions that can extend grid-tied interoperability.

Despite primary research either in optimising turbine's proficiency-effective structural designs or understanding turbulence and wake effects of tidal current profiles, it lacks ascertainment in designing offshore tidal energy conversion system that regulates maximum point power tracking (MPPT) of TST for energy consumers. A power inverter control system for stronger coupling between tidal stream farm and grid's power quality, and developments for offshore DC collection system require more considerations to ensure practical realisation when dealing with large-scale installation for HVDC configured transmissions [7–9].

## 1.1. Literature Reviews and Research Gaps

Customarily, in relation to a direct drive turbine-based energy conversion system, engagement of Back-to-Back (BtB) Voltage Source Converter (VSC), also known as two-staged VSC, serves as a power conversion median that secures synchronisability when coupled to the AC distribution network [10–12]—two mutually connected VSCs with a supercapacitor DC-link connecting in-between. The generator-side VSC is responsible for monitoring MPPT of turbine's active power generations by aligning optimum mechanical speed (closed-loop torque and flux current control) and elevation of blade pitch angles. Contrarily, the grid-side VSC governs a constant DC voltage level at a DC-link, regulates output frequency level for synchronisability with the AC side, and control over the ordering of reactive power from the grid to satisfy voltage ride-through requirements. The following are some selected studies that employ BtB-VSC topology for an energy conversion system using different control approaches.

Harrabi et al. [13] proposed a wind energy conversion system (WECS) that uses BtB-VSC with a Takagi–Sugeno (T–S) fuzzy controller to harvest maximum available power–voltage control algorithm at DC-link. The suggested BtB-VSC employs independent fuzzy models with a decentralised stabilisation approach to govern generator's electromechanical conversion and power quality synchronisation with a grid, respectively. The control principle uses a conventional approach of using a generator's speed feedback to render maximum active power extraction. However, the proposed control algorithm observes operating performance of a single AC drive system. Complications will rise when multiple parallel-connected turbines are linked centrally to a mutual AC-link busbar as disturbance on individual's MPPT and flux current control are forced to be synchronised. Moreover, considerations in ride-through abilities were not incorporated when modelling inverter's control features; hence, its voltage/frequency transient responses toward momentary fault interruptions were not analysed.

Xiaodan et al. [14] developed a cooperative-driven distributed control scheme ideally for active power regulation in a large-scale integration environment. A consensus-based control strategy was proposed to evaluate system-wide performance based on local actions to gain optimal governance of clustered turbines. Likewise, the AC drive system engages a BtB-VSC topology with decentralised ordering of active power ratings to acceleration or deceleration rotor speed independently. However, in view of an aggregated direct drive energy conversion system, adaptation of variable speed control remains essential. It serves as a bandit to mitigate potential loss of active power when dealing with centralised MPPT and a flux current controller. Furthermore, investigations into ride-through transients were not presented as to how the proposed controller compensates during abnormal voltage level events.

Marios et al. [15] propose an optimised offshore tidal current conversion system focusing on formulating control algorithms for pitching regulation against the rotor's variable speed. Similarly, engagement of BtB-VSC topology was engaged to control TST operations for MPPT and synchronisation at

a common AC-link busbar. Indifferently, the authors propose a centralised/shared grid-side VSC to serve a cluster of TSTs while maintaining personal generator-side VSC at respective TST for MPPT operations. Here, the general contribution focuses on building an AC collection system that still uses typical direct torque control features in BtB-VSC for cluster-based TST deployment. However, when converting an AC collection system suitable for HVDC transmission configuration, an additional active rectification stage is required. Despite in-depth investigations into frequency domain and transient responses of TST operations, it lacks operational certainty when TSTs are formatted in a matrix layout having to consume different tidal current profiles (wake and turbulence) at respective tidal channels. How will the control algorithm in generator-side VSC performs for other TSTs in a cluster when securing a constant DC voltage level at a DC collection busbar or resulted in degraded MPPT as field-oriented control (flux current coordinates) is shared across both $d$- and $q$-axis.

From the above-mentioned literature reviews, multi-oriented control frameworks for both separate VSCs can implicate cost infringements, malicious computation due to long distance data transfer, deflation in power qualities, and weak fault ride-through capabilities when enforcing large-scale formation in HVDC-configured transmission. Observations in adopting multiple tidal current channels and MPPT response when using a centralised collection system must be considered when designing a flux current control algorithm. In addition, incorporating voltage and frequency ride-through control features are essential when designing inverters in order to strengthen $PQ$-coupling at a distribution network.

Diversely, Kant et al. [16] proposed a solution that exploits a DC-based direct drive system involving a single generator-side VSC in series with a DC-boost converter before coupling to the DC transmission system. Accordingly, the DC collection system is comprised of parallel connected turbines for large-scale deployment. Uniquely, an energy storage (battery) is installed at the DC-link that links generator-side VSC and a DC-boost converter. It serves as a subsidiary Volt/Watt governor that regulates turbine's MPPT and maintains a constant DC voltage level at HVDC transmission and other dynamic conditions (i.e., harmonic currents, load balancing, and voltage regulation). The results have proved that the proposed Volt/Watt controller can serve as an alternative for reactive power compensation during voltage ride-through support. Likewise, Komal et al. [17] present the use of a two-staged solid state transformer based inverter for a wind energy conversion system that highlights superiority against conventional line frequency transformers.

From the above literature reviews, this paper inspires interoperation of the concept of employing a DC-based direct drive energy conversion system and designing a truncated modular-based DC collection system for HVDC transmission that serves a large-scale offshore tidal stream farm. Moreover, it formulates a control algorithm with respect to inverters to support voltage–frequency ride-through capability and operates in an isochronous mode for stronger coupling with an AC distribution network.

*1.2. Contributions*

The contribution of this paper is twofold:

1.  Modelling of DC Collection Aggregator (DCCA) that serves as the aggregated tidal energy conversion system for multiple parallel-connected direct drive TSTs before coupling to the bi-pole HVDC transmission network. The innovation involves using the sole impedance source inverter (ZSI) that employs solid state transformer technology to support operation variations of clustered TSTs and gains synchronisation at the mutual AC-link (frequency and voltage level). The ZSI's voltage control operation uses a PI controller to secure a constant AC voltage level at AC-link, ordering strategic pulse-width modulation (PWM) signals on switching devices to order a corresponding boosting factor at the impedance source. The DCCA is then coupled to the HVDC transmission along with other TST clusters in parallel to avoid cascading failure when random TSTs experience malfunctions.

2. Design torque-based blade pitch controller that operates close to MPPT to accommodate Volt/Watt function and better generator start-up profile. The purpose of reserving active power generation is to support an under-voltage crisis during low-speed rotor operation (high torque), which is compatible for tidal current physiques.

Two test case scenarios were proposed to validate fault ride-through capabilities. The transient analyses encompass temporary fault interruptions at both sending (offshore)- and receiving (onshore)-end regions and randomly assign TSTs to go offline. The evaluations will capture tidal stream farm synchronisability at the point of common coupling (PCC) with an AC distribution network. Assessments are conducted based on real-time tidal current oscillatory data against the revised fault-ride through standard sanctioned in IEEE 1547-2018.

This paper also extends its contribution into engineering truncated DC collection point and tidal energy conversion system for HVDC transmission from the typical two-stage VSC (BtB-VSC) to a single-stage ZSI with a centralised active rectifier at a respective cluster.

The remaining paper is organised as follows: Section 2 models the proposed 90 MW TST farm in grid-tied engagements, highlighting components in the DC collection system, and employment of a 3-phase dual-active bridge converter using solid state transformer concept development. Section 3 defines control strategies at respective power converters to concede isochronous mode operation, hosting power quality requirements due to softer grid coupling. Section 4 investigates implications on engaging different cluster sizes and capital investment benefits. Section 5 reviews operative results engaging large-scaled TST using real-time tidal current magnitudes data against grid-code compliances and industrial compatibility. Finally, the paper concludes in Section 6.

## 2. Proposed Testbed System: HVDC Transmission for Offshore 90 MW Tidal Stream Farm

The proposed 90 MW TST farm shown in Figure 1 is modelled in PSCAD, servicing 10 identical TST clusters with each having six aggregated direct drive TSTs connected to a common DCCA installed on the offshore converter platform. The parallel connected TST clusters are then coupled to the bi-pole HVDC transmission rated at $\pm 80$ kV *DC* transporting maximum active power generation from offshore to onshore. The engaged TSTs are designed based on *AR*1500 specifications manufactured by Atlantis Resources [18] and deployed approximately 200 km from shore at 30 m below sea level. For each cluster, TSTs are strategically positioned in a staggered 3 by 2 array-geometrical formation with equidistant spacing of 500 m apart from each other. The subsequent TST clusters will then be aligned adjacent to each other. Consequently, adaptations of 3-channel/tunnel tidal stream current profiles will be implemented to discern tidal dissipation due to wakes and turbulences emigrating from front to rear TSTs. The tidal stream farm is divided into two regions separated by the 200 km ABB Submarine Cables [19], sending-end (offshore) and receiving-end (onshore) as seen in Figure 1. The tidal stream farm is coupled at PCC connecting to the AC distribution network, short-circuit ratio of 4, and $X/R = 10$.

The sending-end region is comprised of 10 identical DCCAs, individually programmed to command strategic tidal energy conversion operations for respective direct drive TSTs. Together with the proposed blade pitch controller, the turbines are governed by an over-voltage limiter protection and coordinate maximum electrical torque to achieve desired MPPT against arbitrary tidal speed profile. On the contrary, the receiving-end region is designed to avow interoperability with the 22 kV *AC* medium-voltage AC distribution network using a field oriented controlled single-stage VSC.

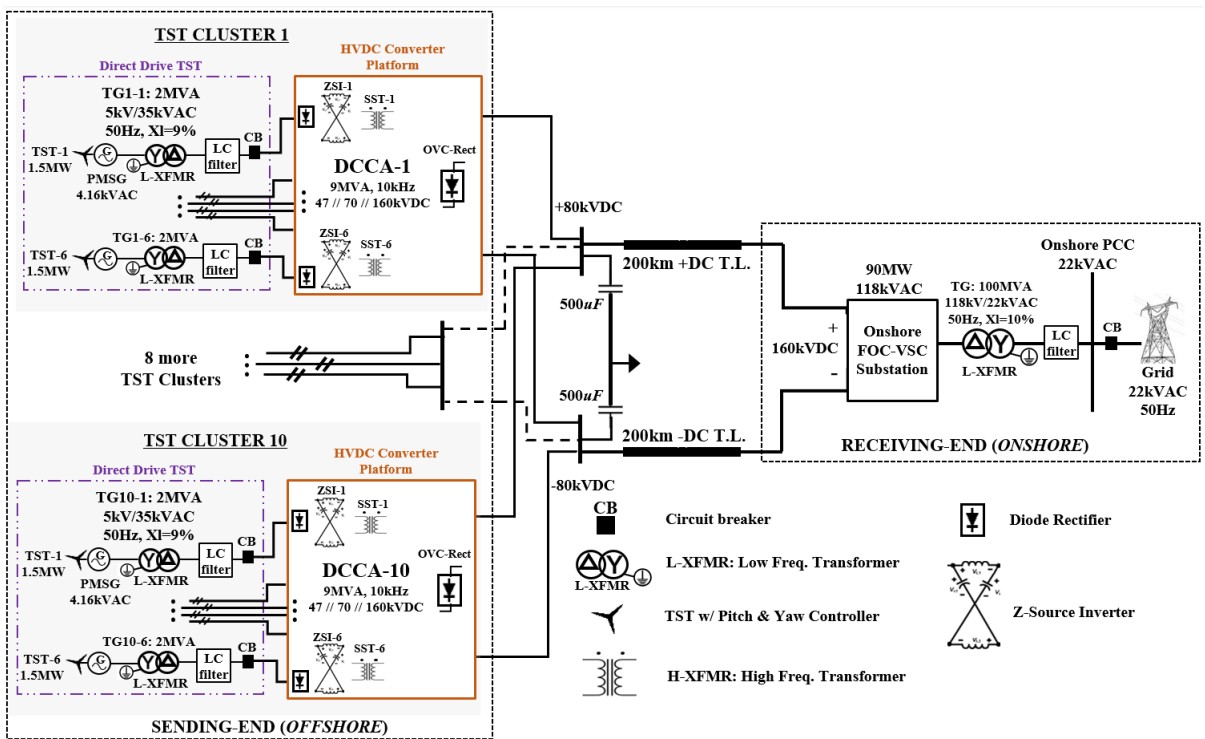

**Figure 1.** Proposed HVDC transmission testbed system for a 90 MW offshore tidal stream farm.

### 2.1. Tidal Stream Turbine

Characteristically, the physical dynamic operations of a TST is very comparable to a wind turbine, considering both have correlative design physiques and MPPT control strategies [20]. Based on *AR*1500 TST specifications and real-time tidal current speed data, the turbine's power coefficient administrations can be profiled into a look-up table to pursue MPPT at different influential parameters (pitch angle versus rotor speed). The MPPT performances showed that torque feedback regulator (Watt/Speed) is required in addition to the speed controller. This infuses high torque inducements to gain low mechanical speed where proportional active power generation is actualised based on PMSG's magnetic field using a power–torque–speed relationship. In this sense, despite typical speed-frequency function for TST MPPT, the proposed control architecture uses blade pitching to transcend electrical torque biasness which forces rotor speed to operate below 1.0 p.u. hence de-synchronism in generator's output voltage level at mutual AC-link. However, such crisis can be resolved by the tidal energy conversion system (DCCA) where its voltage boost controller provides constant DC voltage support at HVDC transmission.

Supplementarily, to support PMSG's high torque operations and voltage ride-through ability, the blade pitch controller also exploits Volt/Watt control as the secondary dynamic voltage support. Given that the TST only observes MPPT operation, it lacks in directing reactive power compensation during voltage level deviations at PMSG. Consequently, the blade pitching control mechanism (Volt/Watt function) observes close to MPPT operation which reserves spinning active power generation to curb over- or under-voltage crises, which increase blade pitch angle when there is over-voltage and vice versa.

### 2.2. Uncertainty and Disturbance Affecting Tidal Stream Turbine Operation

Realisation in tidal stream current uncertainty requires special attention as an initial classification of a high energy tidal current site is often defined by its peak velocity and range of water depth. However, peak current velocity fails to provide an accurate expression of potential power production involving

fine scale temporal and spatial variability in tidal energy flow [21]. Furthermore, simulation assessments in tidal energy conversion assume that the array of TSTs are operated with instantaneous/similar tidal flow intake, which renders false transient analyses especially in cluster formation. The characteristics of tidal current cycle composites of both flood and ebb phases arbitrate unsymmetrical flow responses in nature. Thus, explained and proposed in [22], modelling of turbine parametrisation uses combination effects of asymmetry tidal and yawing misalignment disturbances before estimating available potential energy harvested, *P*. It uses a function of undisturbed depth-averaged tidal resource velocity defined as:

$$P = \begin{cases} 0, & |V cos^{1/3}(\gamma)| < V_{ci} \\ 0.5\rho V^3 cos^\beta(\gamma) A C_p, & V_{ci} \leq |V cos^{1/3}(\gamma)| \leq V_r \\ 0.5\rho V^3 cos^\beta(\gamma) A C_p, & |V cos^{1/3}(\gamma)| > V_r \end{cases} \tag{1}$$

where *A* represents the rotor blade sweeping area, 1017 m$^2$, and $\rho$ is the water density rated at 1000 kg/m$^3$. $V$, $V_{ci}$, and $V_r$ denote the instantaneous tidal current velocity, cut-in velocity (3 kn), and rated velocity (4.9 kn), respectively. $C_p$ dictates the turbine's power coefficient increasing from 0.07 to 0.496, which incorporates the effects on losses (i.e., power train efficiency, turbulent effects, tip-speed ratio, and rotor performance). $\beta$ defines the yaw misalignment factor ($\beta = 2$) in relations to marine turbine deployments.

### 2.3. Tidal Stream Turbine Generator Set

A permanent magnet synchronous generator (PMSG) was selected as the generating unit for TST due to its self-excited system which has high-energy density magnets that intensify rotating magnetic field without the use of conventional rotor winding. Therefore, low-speed and gearless applications are compatible when engaging a direct drive TST system. Decisively, PMSGs pose operational appetencies towards gratifying low maintenance costs, long lasting winding insulation life length, avowing of low operating frequencies, and lightweight quality. Discussed in [10,11,15], they described PMSGs' three stator winding voltages in direct-quadrature-zero ($DQ0$) reference frame to control MPPT operations.

### 2.4. DC Collection Aggregator: Connecting Clustered Direct Drive TSTs to HVDC Transmission

To replace engagement of a two-stage VSC (BtB-VSC) topology for a turbine's tidal energy conversion system, an original DCCA seen in Figure 2a was proposed to govern a cluster of six TSTs. It apprehends TSTs' voltage and speed variations, and secures a constant DC voltage level with high active power transference at HVDC transmission. The individual AC-DC passive rectifier which couples to respective TST is connected to a ZSI and high frequency solid state transformer (H-XFMR) before aggregating with other TSTs to a mutual 118 kV AC-link. The AC-link carries a levelled voltage and frequency based on the ordered parameters, synchronising TSTs' performance attributes despite their disassociated operations. ZSI is habitually immune to electromagnetic interference and voltage surges due to their primary control in regulating boost voltage level oriented by the shoot-through operation mode. Uniquely, ZSI has the ability to simultaneously function as a boost converter at the impedance source and convert DC–AC voltage (inverter). Hence, customary inheritance of frequency and voltage droop controls is naturally integrated to provide isochronous operation mode. Subsequently, to accommodate high frequency order ZSI operations, H-XFMR is engaged. Indeed, there are limitations when designing H-XFMR to operate at desired parameters–higher flux density tolerance and saturation region, large magnetic losses, and heat dissipations, and it is not technological and industrial applicable intended for large power ratings ($>MW$). Therefore, this paper avoids placing the mutual AC-link busbar right after ZSIs and deploys a common H-XFMR even though it reduces greater installation costs. Finally, a centralised over-voltage control rectifier (OVC-Rect) is employed before coupling to the HVDC transmission. The improvised

OVC-Rect is comprised of an LCL-configured AC filter connecting to a three-phase diode rectifier with a voltage chopper at DC-link. The voltage chopper serves as a secondary controller that uses a half-second delay Volt/Watt function to manipulate a DC voltage level at HVDC transmission—decreasing active power when voltage swells and vice versa through strategic ordering switching sequences.

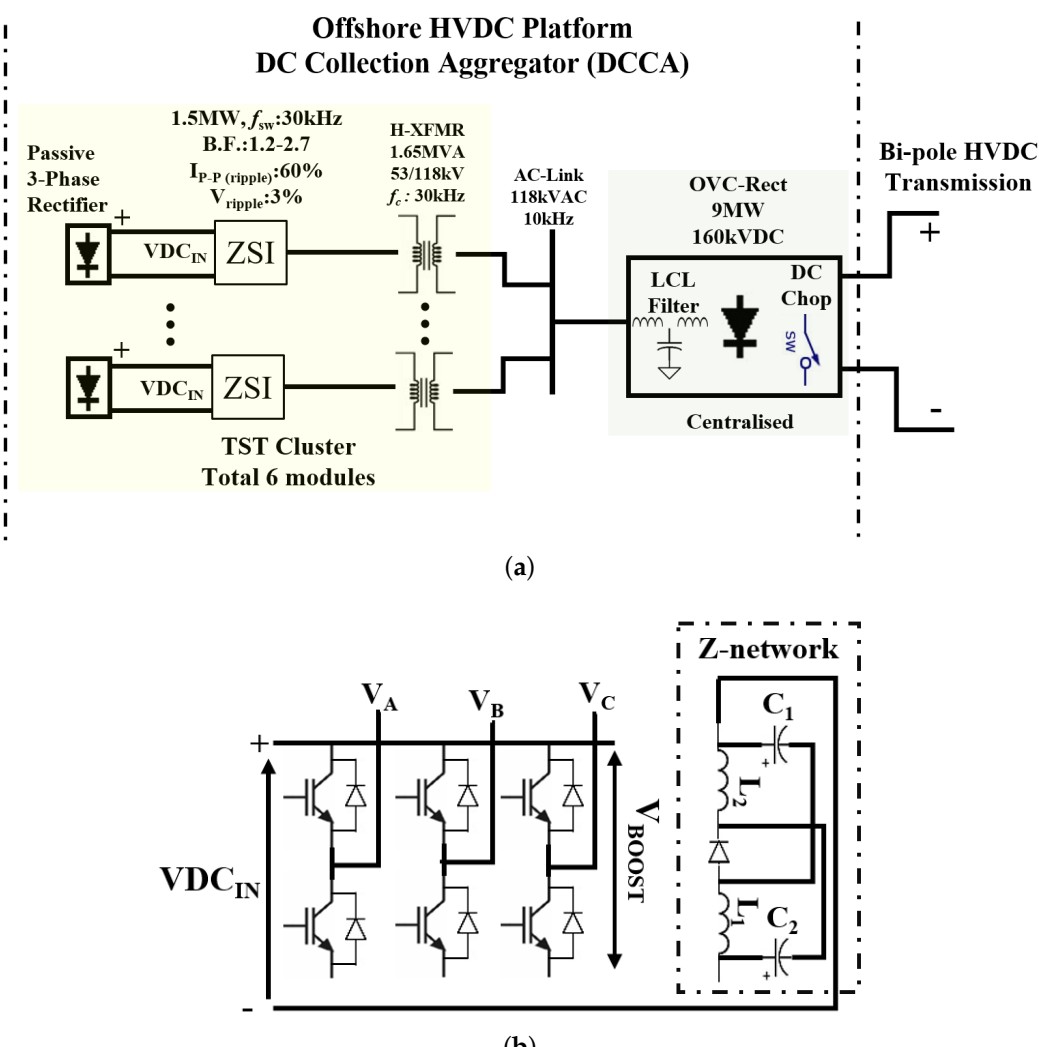

**Figure 2.** (**a**) single DCCA module serving six direct drive TSTs for tidal energy conversion system; (**b**) Impedance Source Inverter (ZSI) electrical layout.

Figure 2b exhibits ZSI's electrical design that is comprised of two interdependent operations: a shoot-through (short-circuiting) abled DC–AC inverter, and an impedance source network (Z-network). The two-port Z-network inherits split inductors, *L*1 and *L*2 and cross-connected capacitors, *C*1 and *C*2 that can be expressed as an energy storage bank with a second-order filtering element. The operation of switching gates in the inverter uses shoot-through gating sequences where two switching devices in the same line are gated concurrently to orchestrate a controlled DC voltage surge, creating voltage boost engagements. Typically, the Z-network is configured as input for the inverter circuit; however, the proposed improved ZSI suggests that the Z-network will be serially-connected after the inverter and closes the circuitry back to the input. Such configuration proves to suppress startup inrush current

and stabilise resonance between Z-capacitor (i.e., excessive voltage) and Z-inductor (i.e., current ripples). Detailed proof of lemma for ZSI operations are in Appendix A.

### 2.5. High Frequency Transformer

The design process of proposed DCCA employs an H-XFMR operating at 30 kHz ($f_c$ carrier frequency), serving as a voltage step-up between ZSI and OVC-Rect to match $\pm 80$ kVDC at HVDC transmission. Unfortunately, in relation to market and industrial availability, deploying of H-XFMR rated beyond 10 kHz for more than $10^2$ Kilo-Watt applications are still premature yet alone operating at 9 MVA of power loading capacity. It forces the flux density to operate in a saturated region causing inductance to collapse and thus large magnetic losses and heat dissipations will be impinged. Feasibility in producing high magnetic field strength while securing a large permissible flux density region is still inaccessible based on today's magnetic core material technology breakthrough.

Nevertheless, researchers have proved theories in initiating state-of-the-art H-XFMR prototypes (i.e., >20 kHz) by materialising fusion magnetic core materials that proffer higher flux density tolerance and saturation region (i.e., $B_{sat}[mT] > 1450$) while provisioning large power density (i.e., 10 kVA < P < 500 kVA) with low core losses [23–27]. Table 1 depicts aggrandising technological trends in H-XFMR interests to condone rapid transitioning of SST technologies for power converters. However, in relation to the proposed H-XFMR rated at $9MVA$, 10 kHz, 1:2.5, it has yet to find an accommodative solution. Even so, the future still holds a promising breakthrough in finding a suitable Nano-Composite Magnet Technology for high frequency Mega-Watt scaled power converters.

Alternatively, the DCCA configuration can be decomposed into lower power ratings (i.e., decrease current carrying capacity) by cascading multiple ZSI with independent H-XFMR in parallel to respective TST as shown in Figure 3 based on available H-XFMR shown in Table 1. Nevertherless, such implementation does incur high manufacturing costs where power conversion devices will aggrandise significantly alongside its control architecture. Logically, such preposition is not justifiable when mitigating flux density saturation constraints against cost increment; thus, this paper assumes a centralised H-XFMR was engaged based on proposed ratings for each cluster as seen in Figure 2a.

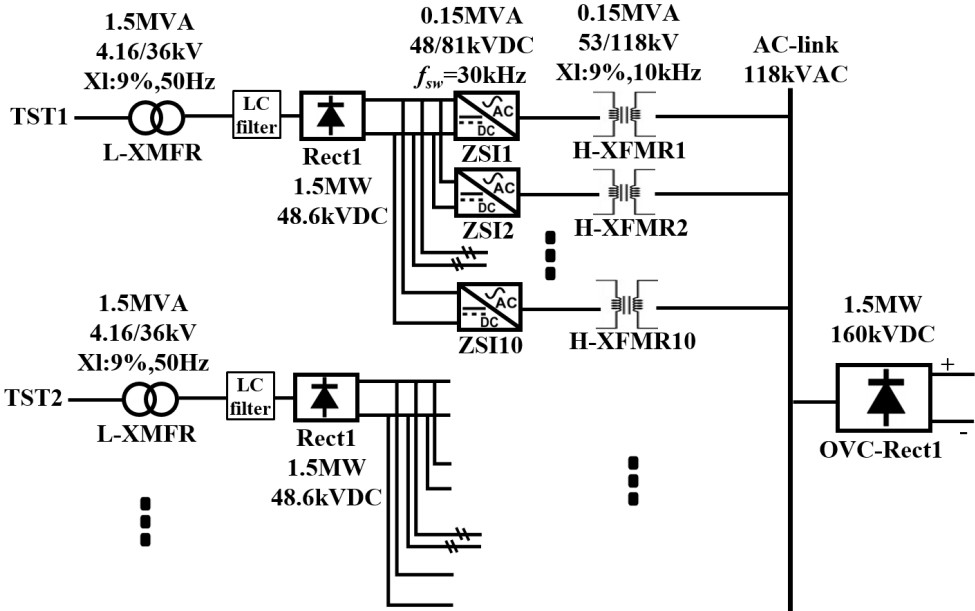

**Figure 3.** Alternative DCCA configuration for individual TST based on industrial available H-XFMR ratings.

**Table 1.** Technological advancements in high frequency transformer.

| Ref. Year | Magnetic: Material | Core Series | $B_{sat}$ @ 100 °C | Power Rating | $V_{PRI}/V_{SEC}$ | Freq. |
|---|---|---|---|---|---|---|
| [23] 2013 | Finemet: FT-3H | UU, EE | 0.13T | 2 kVA-250 kVA | 380 V/5 kV | 20 kHz |
| [24] 2015 | Ferrite: N87 | EE | 0.118T | 5 kW | 215 V/344 V | 50 kHz |
| [25] 2016 | NanoCrys.: FT-3M | UU | 0.8 T | 30 kVA | 0.7 kV/64 kV | 20 kHz |
| [26] 2017 | Metglas: 2605SA1 | CC | 0.55 T | 166 kW | 400 V/1 kV | 20 kHz |
| [27] 2018 | Ferrite: 3F36 | EE | 0.118 T | 15 kW | 0.4 kV/4.16 kV | 0.5 MHz |

*2.6. Submarine Power Cable (Single Core)*

Subsea HVDC transmission technologies have proved to be the most economical solution in bridging offshore generation systems into mainland electrical network by a single point-to-point connection. Cost effectiveness in engaging DC cables can be optimised to curb expensive installation costs of HVDC converter stations onshore against the AC transmission system. Regardless, with proper valuations of comparative cost information, attractive return on investments can be accomplished [28]. Dual DC single-core copper conductor cables manufactured by ABB HVDC Light shown in Figure 4 are modelled for the testbed system.

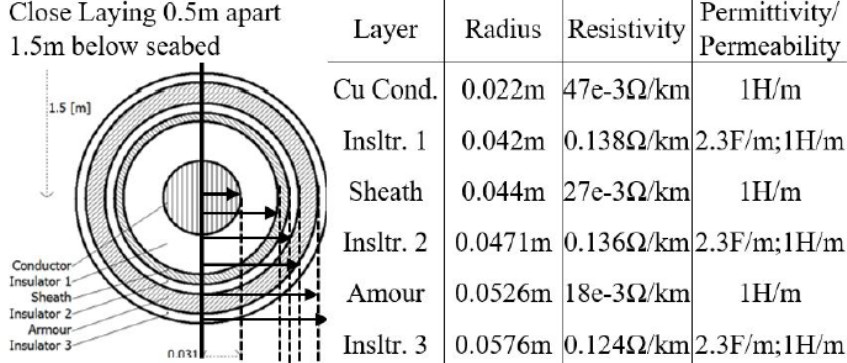

**Figure 4.** 200 km close laying ABB submarine cable model.

**3. Proposed Control Algorithms**

*3.1. Turbine Dynamic Blade Pitch and Yaw Controller*

Two affiliated control schemes were proposed to secure MPPT of direct drive TST operations: hub yawing manoeuvring in relation to tidal current migrations and blade pitching in response to tidal speed with the constraint of PMSG speed boundaries.

Figure 5a presents the turbine's yawing control scheme in response to the migrations of tidal current angle of attack against the turbine's hub orientation. Tidal current migrations are very predictive based on high and low tide displacement of the moon's gravitational pull. Hence, it is intuitive to pilot the turbine hub to be in a perpendicular direction against the tidal current to gain effective rotor area when harnessing opposite forces. A simple yaw controller can be seen by using a vector approach to calculate $\Delta\theta_{yaw}$ (Coterminal Angles) between tidal current and turbine hub direction. Using a PI controller, the yawing angle is adjusted accordingly based on the angle difference between tidal current and hub orientation, assuming that the turbine's hub has 360° of yawing freedom.

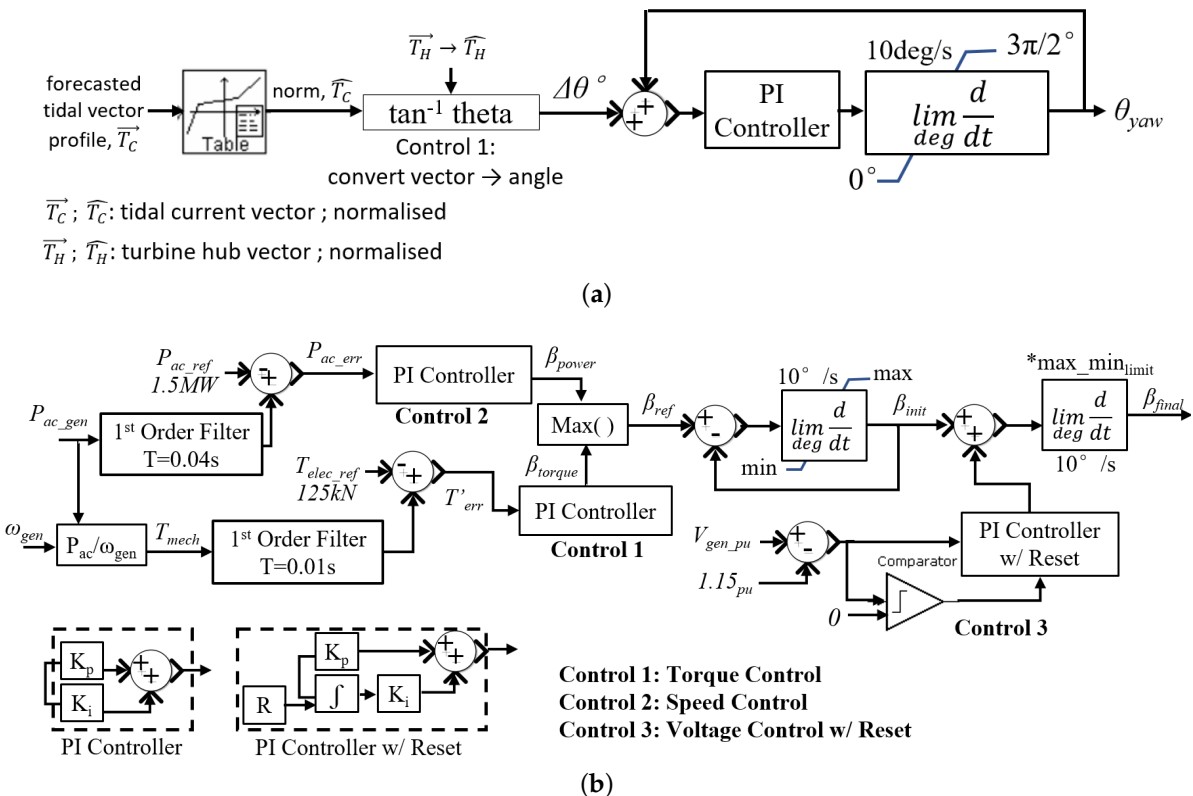

**Figure 5.** (**a**) TST yaw controller; (**b**) TST torque-controlled blade pitch controller.

Subsequently, the blade pitch controller has two operating objectives in gaining optimal MPPT and dynamic voltage regulatory at PMSG. Primarily, the control algorithm is based on Watt/Speed function to achieve MPPT despite having high-torque/low-speed responses when operating in an aggregated formation. Thus, Figure 5b suggests a blade pitch controller that incorporates torque feedback control to dictate a repressed mechanical speed limit at PMSG (i.e., <1 p.u.). As a result, the PMSG's voltage rating will be regulated below rated value, affecting the turbine rectified output voltage to be levelled between 0.4–1.0 p.u. regardless of tidal speed profiles. Here, the pitch angle is restricted only to a 70% degree of freedom and PMSG's reference/nominal speed is capped at 10% lower than the rated specification. Consequently, the cut-out speed of the turbine will decrease and the blade angle will be not levelled at 0° (i.e., initial pitch > 0°) when operated at nominal speed. In addition, Volt/Watt function was added to the blade pitch controller to serve as a secondary over-voltage regulator especially during PMSG start-up where the active power is decreased at TST; forcing power injection from the grid. In this sense, the remaining 30% pitching angle was reserved to accommodate over-voltage crises by increasing pitch angle, which in turn decreases active power generation. For Volt/Watt function, a PI controller with reset ability was employed to ensure that reset activation occurs only during over-voltage crises.

### 3.2. DC Collection Aggregator (DCCA) Controller

The proposed control architecture in DCCA composites of two electrical regions separated by the H-XFMR: (i) rectified direct drive TST coupled to ZSI governing TST's tidal energy conversion system with voltage ride-through compliance, and (ii) a common/centralised Freq/Watt controlled rectifier, OVC-Rect

that services as an frequency- and over-voltage regulator at HVDC transmission. In addition, the ZSI is designed to improve start-up ramp profiles to prevent inrush current and sudden voltage dip.

The ZSI control framework maintains a constant voltage level of 118 kV *AC* and frequency of 10 kHz (refer Figure 2a) at mutual AC-link despite mechanical speed variations generated at respective TST. The control proceedings employ a PI-based voltage loop controller topped with the ZSI shoot-through operation mode to gain a constant DC voltage level at HVDC transmission regardless of unregulated frequency and voltage profile arbitrated from TST during MPPT. Positively, ZSI forgoes the concept on using a current loop controller as it exploits a unique voltage boosting operations that allow fast response to load variations and avoid loop gain when varied with input. Shown in Figure 6a, the ZSI controller involves three control theorems: (i) Calculate suitable ratings for the passive devices in the Z-network using tolerance boundary approach to gain sufficient boosting ability. (ii) Implementing AC Voltage Controller to regulate desirable and constant AC voltage capacity at AC-link and determining its value to serve as a feedback reference voltage. (iii) Implementing DC Boost Feedback Controller, directing the six-pulse PWM signals for respective switching devices to create controlled surge voltage that charges Z-network to perform boosting process. The control algorithm sequences are defined as follows:

(i) Formulations in determining Z-capacitor and Z-inductor ratings in the Z-network are strictly dependent on the required Boosting Factor (BF) and inverter's switching frequency when generating desirable DC boost voltage level (average), $Vdc_{boost}$. However, it is unrealistic to alter passive component values in real time to preserve constant $Vdc_{boost}$ as BF will deviate constantly in response to the uncontrolled DC voltage generated at TST, $Vdc_{WT}$. Nevertheless, through investigations, a trend was observed on the Z-network's efficiency where it is permissible to retain inductance and capacitance values without having $Vdc_{boost}$ level be clipped. The key is to identify BF limits against the available $Vdc_{WT}$ region/range before finalising the appropriate AC voltage at AC-link as a reference voltage:

$$L_1 = L_2 = \frac{B.F. - 1}{f_{sw} \times 2B.F} \times \frac{Vdc_{WT} + Vdcboost}{2\Delta I_{L_{max}(pk-pk)}}$$

$$C_1 = C_2 = \frac{B.F. - 1}{f_{sw} \times 2B.F} \times$$

$$\frac{2Idc_{WT}}{(Vdc_{WT} + Vdc_{boost}) \times ripple\%} \tag{2}$$

*s.t.* following constraints:

$$1 \leq B.F._{min,max} \leq 3$$
$$29.5kHz \leq f_{sw} \leq 30.5kHz$$
$$1.01 \times Vdc_{WT} \leq \Delta V_{C_{pk-pk}} \leq 1.03 \times Vdc_{WT} \tag{3}$$
$$0.88 \times Idc_{WT} \leq \Delta I_{L_{pk-pk}} \leq 1.16 \times Idc_{WT}$$

(ii) Strategical selection of AC voltage level as reference is required to avoid voltage divergence at AC-link, $Vac\_link_{L\text{-}L}$. Label (4) formulates fitting AC voltage level at AC-link taking the consideration of 3% voltage ripple during the shoot-through operation. In addition, it was considered to take the minimum BF when assigning the reference AC voltage in order to anticipate $Vac_{WT}$ shortfall during instances where WT's mechanical speed decreases due to low wind speed input.

$$Vac\_link_{(L\text{-}L)} = \frac{\pi \times \sqrt{2} \times BF_{min} \times Vdc_{WT}}{6} \times 0.97 \tag{4}$$

$$Vdc_{ave} = \frac{Vac_{L\text{-}L} \times 3 \times \sqrt{2}}{\pi} \tag{5}$$

(iii)   The DC Boost Feedback Controller was designed to direct 6-pulse PWM signals ordained by BF in real time. It influences the shoot-through period given in Label (A6). Subsequently, engaging Label (A7), the updated PWM signals were generated provoking the switching operations of the inverter to create voltage surge that charges up the Z-network for boosting proceedings.

(iv)   It can be seen that the AC Voltage Controller uses an adaptive PI compensator that uses back-calculation algorithm to prevent continual increment of input. Thus, in the linear range, the integrated error and the difference between saturated and unsaturated signals serves as feedback that controls the integral state in the saturation region defined in Label (6). Here, constant AC-Link voltage level is secured by ordering required $Vdc_{boost}$ magnitude through BF manipulations based on capricious $VDC_{IN}$ ratings:

$$q = \begin{cases} e, & if\ u = v \\ e - k_a(u - v), & if\ u \neq v \end{cases} \tag{6}$$

When employing a back-calculation algorithm, accidental reset of integrator due to input saturation caused by malicious interpretation of error is alleviated. The integral state will be reduced along with its time constant when PI controller's output gets saturated. It provides negative feedback from the controller's output, else the integral state accrues the AC voltage error and initiates conventional PI proceeding.

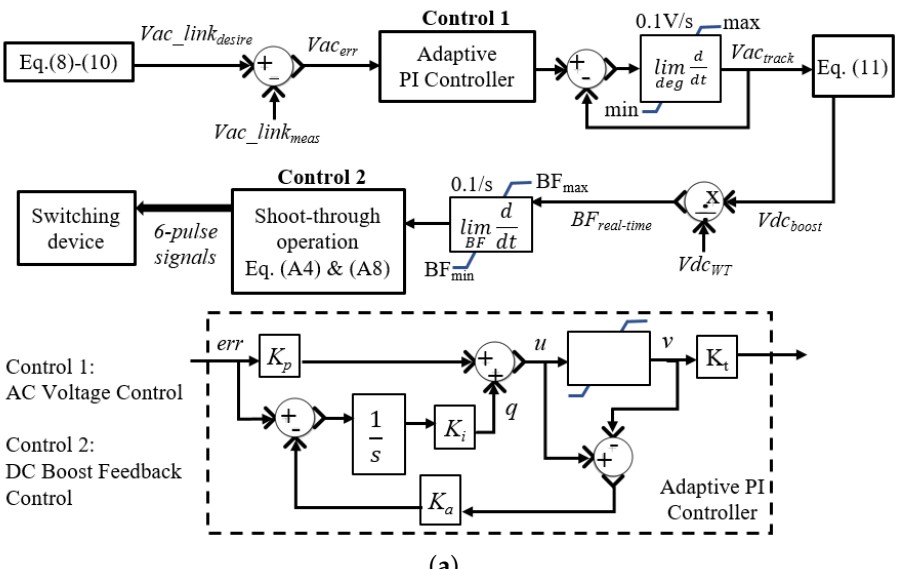

(**a**)

**Figure 6.** *Cont.*

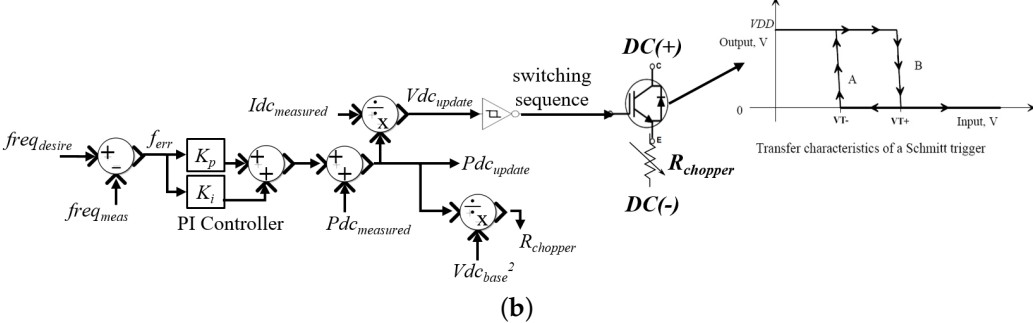

**(b)**

**Figure 6.** (**a**) control algorithm of ZSI; (**b**) OVC-Rect controller with Freq/Watt functionality.

Contrarily, the OVC-Rect controller seen in Figure 6b was designed solely as a frequency ride-through precaution in the case when secondary-side of H-XFMR induced with momentary fault. Through investigations, the measured frequency level will increment at every fault incident and will not regain normality. Thus, the diode rectifier was modified with a DC Chopper controller that supports Freq/Watt function to order decremental DC voltage level to decrease active power generation which in turn decreases over-frequencies crises.

### 3.3. Grid-Side Voltage Source Converter Controller

The grid-side VSC controller ensures that full interoperability is achieved between tidal stream farm and grid (i.e., Frequency, AC voltage, receiving-end HVDC transmission). The controller mimics a typical VSC control [29,30] that proffers frequency and active power regulatory shown in Figure 7. The transference of active power in relation to the regulated DC voltage level was based on manipulating $q$-axis component, $M_q$. Diversely, for the $d$-axis component, employment of a feedback controller was utilised to maintain a synchronised 22 kVAC, 50 Hz at grid by attuning $M_d$. To determine position of the rotating coordinate frame, $\theta_i$, it uses phase lock loop (PLL) function and measured AC voltage at grid.

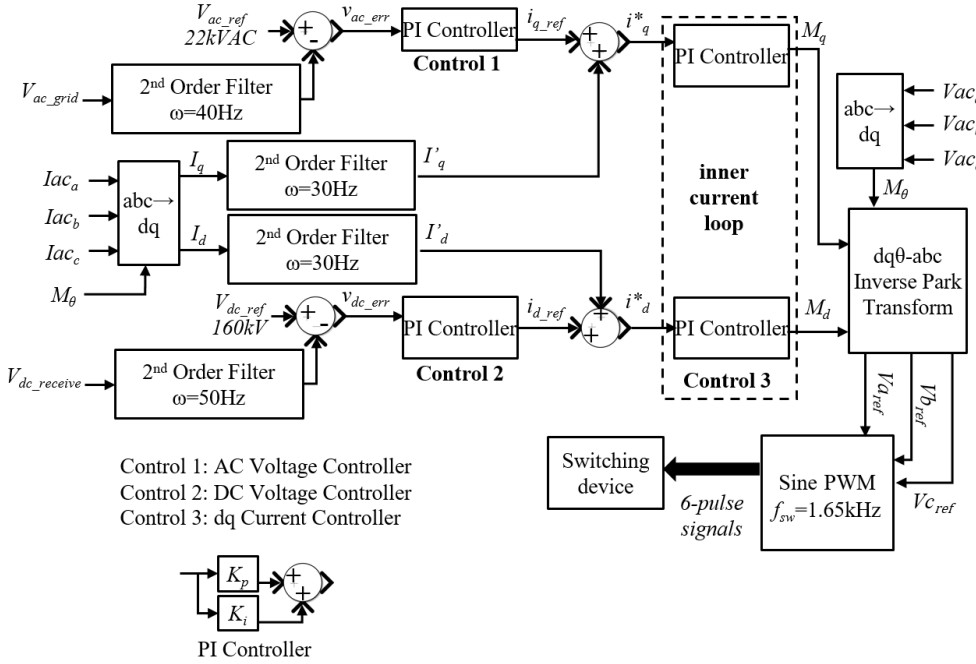

**Figure 7.** Control architecture for Grid-side VSC.

## 4. Cluster Sizing and Capital Investments

### 4.1. Inferences in Adopting Large Scale TST Operations

In [31], a larger power coefficient index ($Cp$ > theoretical limit of 0.592) is reaped when TSTs are clustered and deployed along a single tidal channel that has low input tidal current. However, when multiple tidal channels are mediated, assenting a levelised loading of power generations from all clustered TST are illicit. Factors such as momentum fluxes, and migration delays can influence retardations of the tidal current flow at downstream turbines. Thus, turbulences and linear delay elements at consecutive tidal channels are induced to impart the effects on free-stream tidal flow reductions.

To further augment realisticity, implications of tidal current dispersions using Weibull distribution expressed in (7) are introduced where dimensionless Weibull shape parameter, $\alpha = 2$ (Rayleigh distribution), and scale parameter of $\beta = 8.8$ kn functions are adopted. They correlate with an average tidal current velocity of 8.363 kn that results in a tidal stream farm capacity factor (ratio of power over theoretical energy generated at a rated wind speed) of 40%:

$$f(V_{tc}; \alpha, \lambda) = \frac{\alpha}{\beta} \times [(\frac{V_{tc}}{\beta})^{\alpha-1} \exp^{-(\frac{V_{tc}}{\beta})^{\alpha}}] \tag{7}$$

When cooperating both tidal current delay responses and Weibull distributions, energy losses were expected to conceived at a different tidal channel. The induced voltage influences PMSG's performances at lower operating speed causing tidal current of other channels to experience high torque but low power generation. Thus, Table 2a exposes DCCA's efficiency and turbine's mechanical given at different cluster sizes and specify time delay against tidal current. A trend can be deduced that employing cluster size of six aggregated TSTs is ideal to gain levelised efficiency performance given at PMSG's speed displacements. Despite having two aggregated TSTs rendering at higher efficiency, it fails to uphold when infiltrating below a cut-in speed region. Likewise, the eight aggregated TSTs present a small deviation factor for PMSG speed and efficiency; however, it surfaces deployment constraints, especially for H-XFMR application.

**Table 2.** (**a**) Dynamic influence on TST aggregate size for a single DCCA; (**b**) comparisons of installation costs' breakdown.

| (a) | | | | |
|---|---|---|---|---|
| Tidal Speed | 2.8 kn | 3.51 kn | 5.78 kn | 7.86 kn |
| DCCA configured with 2 Aggregated TSTs | | | | |
| Efficiency (%) | 89.08 | 91.48 | 94.78 | 95.49 |
| PMSG Speed (p.u.) | 0.388 | 0.794 | 1.002 | 0.996 |
| DCCA configured with 6 Aggregated TSTs | | | | |
| Efficiency (%) | 92.26 | 92.84 | 93.21 | 93.53 |
| PMSG Speed (p.u.) | 0.531 | 0.711 | 0.852 | 0.941 |
| DCCA configured with 8 Aggregated TSTs | | | | |
| Efficiency (%) | 91.36 | 91.43 | 91.75 | 91.14 |
| PMSG Speed (p.u.) | 0.521 | 0.592 | 0.716 | 0.893 |

| (b) | | |
|---|---|---|
| **Large Scale Deployment** | **MCT** | **Proposed** |
| | **SeaGen [32]** | **TST** |
| Power converter system (Freq. conv.) | 894 $/kW | 798 $/kW |
| Onshore power grid interconnections | 50 $/kW | 119 $/kW |
| Control system per TST | 13 $/kW | 9.5 $/kW |
| Annual Operation and Management Cost | 99 $/kW | 87 $/kW |
| Total savings | − | 42.5 $/kW |
| Cost of energy (cents/kWh) | 4–6.5 | 3.56–5.2 |

*4.2. Cost Benefits' Comparisons*

The proposed DC collection system inflicts further cost reductions for large-scale offshore deployments. With the new inexpensive tidal energy conversion system and dynamic active power harvestation controller at individual TST, coordination in governing the generator-side of HVDC topology has greatly reduced in costs. Table 2b summarizes the investment breakdown costs in deploying a medium-scaled tidal stream turbine farm (12 turbines). Comparisons were made between Marine Current Turbines (MCT) SeaGen [32] against the proposed DC-based TST topology.

## 5. Simulation Results

A nonlinear simulation environment using PSCAD was engaged to validate the proposed tidal stream farm testbed power quality performances and transient responses with respect to ride-through functions. Selected test case scenarios were imposed to view controllers' responses towards tidal energy conversion system optimality and coupling strength (power quality) at PCC. The objective is to procure maximum power deliverances while guarding tidal stream farm interoperability with the AC distribution network. In addition, obligations in complying ride-through standards are considered to ensure uninterrupted and reliable operation of TSTs. Together with the data specifications presented in Appendix A, selected analytical studies evaluating system's performances and transient responses are as follows:

(i)   *DCCA in governing tidal energy conversion system*—Transient and performance analyses, inspecting frequency and voltage qualities in directing direct drive PMSG variable against a real-time tidal current profile [33]. Conjointly, analyses on the proposed adaptive PI controller in ZSI and blade pitch controller were presented governing prescribed ratings at DCCA's AC-Link busbar.

(ii)  *Transient at PCC coupled to the 22 kVAC distribution network*—Here, investigations were focused on the HVDC transmission, both sending- and receiving-ends in relations to power delivery administrations and satisfying ride-through capability along with dynamic voltage support requirements. The analyses then advances towards the grid-side VSC in legislating synchronicity with the 22 kVAC power grid.

(iii) *Random Offline TSTs*—Investigates transient responses and grid synchronicity quality when random TSTs go offline.

*5.1. Evaluating Single TST Operations Based on Proposed Control Strategy in DCCA*

Pre-analytical investigations of a single TST performances were reviewed. It verifies that the blade pitching and torque feedback (Watt/Speed function) controllers can procure high torque-low speed operations while curbing a potential over-voltage phenomenon rendered by Volt/Watt function to heighten ride-through capability. Operation of TST is separated into two analyses: controlled and arbitrary tidal current speed profile as input.

5.1.1. Torque-Controlled Blade Pitching and Yawing Administrations: Constant Tidal Speed

Results presented in Table 3 depict TST's operations based on specified tidal current speed (i.e., from cut-in to cut-out). Monitoring of PMSG's mechanical speed and torque, active power generation, and blade pitching elevations were recorded to apprehend operation characteristics influenced by the proposed TST's control design. As expected, the torque feedback controller has successfully dominated high active power generation, thus suppressing mechanical speed to gain greater torque jurisdiction. Consequently, the blade pitch angle was levelled at $5.96°$ when operating at cut-in tidal speed due to intentional augmentation of PMSG's nominal mechanical speed when designing the controller as compared to manufacturer's specification. Such scheme annex hindrance on maximising TST's original

cut-out tidal speed limit concerning over-voltage operation is due to high mechanical speed reference. To apprehend over-voltage crises, the Volt/Watt function is activated through higher blade pitching angle; thus, the maximum pitching of turbine's blade was rated at 18.82° instead of 28° at cut-out speed.

**Table 3.** PMSG performances (Std. Dev. 0.516).

| Tidal Speed | 3.50 kn | 4.76 kn | 5.89 kn | 8.15 kn |
|---|---|---|---|---|
| $V_{PMSG}$, $(p.u.)$ | 0.408 | 0.875 | 0.989 | 0.992 |
| $I_{PMSG}$, $(Arms)$ | 150.61 | 372.80 | 364.83 | 363.242 |
| $P_{PMSG}$, $(MW)$ | 0.302 | 1.357 | 1.501 | 1.499 |
| $Q_{PMSG}$, $(MVar)$ | 0.0625 | 0.55 | 0.55 | 0.55 |
| $\omega_{PMSG}$, $(p.u.)$ | 0.411 | 0.869 | 0.956 | 0.963 |
| $\tau_{PMSG}$, $(p.u.)$ | 0.482 | 0.914 | 1.007 | 0.998 |
| $C_P$ | 0.496 | 0.338 | 0.164 | 0.072 |
| Pitch angle,deg | 5.96 | 9.42 | 17.53 | 18.82 |

### 5.1.2. Torque-Controlled Blade Pitching and Yawing Administrations: Arbitrary Tidal Speed Profile

Investigation on the dynamic voltage support during TST operations can be viewed based on the 12-hr arbitrary tidal current profile recorded at Beazley Passage of West Coast Vancouver Island shown in Figure 8a. It also demonstrates transitional migration of tidal current from high tide to low during 40 s to 55 s of simulation time.

Figure 8b illustrates TST performances in response to the proposed blade pitching control schemes to deal with $\omega_{PMSG}$ and $\tau_{PMSG}$ relationship. Notice that there were instances where the Volt/Watt function was activated and successfully suppressed over-voltage crises, capping voltage level at 1 p.u. while pitching $\theta_{yaw}$ more than 20° to decrease $P_{PMSG}$. During the transition of tidal direction, the TST's rectified DC voltage did not dip instantaneously due to large capacitor discharge behaviour at DC-link.

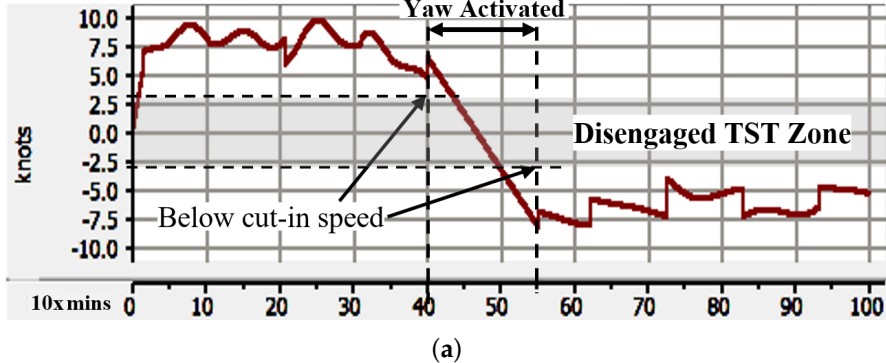

(a)

**Figure 8.** *Cont.*

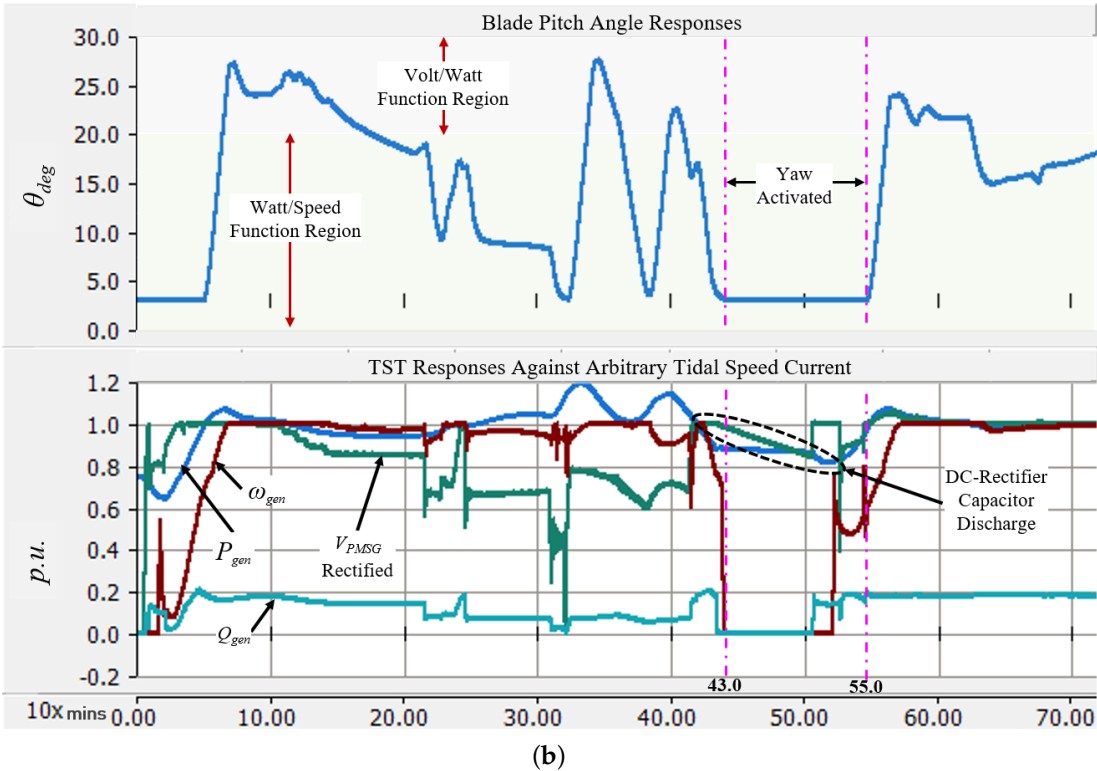

**Figure 8.** (**a**) 12 h (12 pm–12 am) tidal speed profile (*knots*); (**b**) turbine's PMSG performance profiles.

*5.2. Tuning DCCA PI-Controller for Single Direct Drive TST under Arbitrary Tidal Speed Profile: Tidal Energy Conversion System Performances*

Due to tidal current velocity uncertainties, the proposed control proceedings in directing the tidal energy conversion system needs prior tuning in ZSI to preserve responsive feedback in supporting voltage ride-through criterion despite MPPT deviations due to arbitrary tidal speed. Through continual ordering of integral and proportional gain factor using a trial and error method shown in Table 4a, a knowledge-based database is assented to procure best-fitted feedback response serviced by the ZSI in guarding DCCA's AC-Link attributes and Volt/Watt operations. Ultimately, a refined adaptive PI-controller is then conceived by incorporating IF-ELSE decisional statement into the knowledge database to select respective $K_p$ and $K_i$ based on voltage operations generated by PMSG.

Figure 9a(i) illustrates adaptive $K_p$ and $K_i$ directed autonomously in real time, defining respective scalar gain at different time-step to secure constant 58 kV *AC* at ZSI output. Consequently, it minimises error deviations between ordered AC voltage against PMSG's rectified voltage. Figure 9a(ii) exhibits the voltage transient generated at ZSI output which complies diligently against a voltage ride-through criterion where transient falls within the regulatory threshold boundaries labelled in Figure 9a(iii). A minor variance signal-to-noise ratio of 97 dB was propagated due to aggressive alteration of boosting factor in Z-network to coordinate large tidal current digressions. Indeed, its acquisitions can be improved by tightening the damping ratio in the PI controller, administering well-guarded oscillatory responses. However, the system fails to oscillate quickly into a stable trajectory outturning itself towards divergence when responding to extensive deviations of input signal. Nevertheless, the overall TST performances have shown positive operative margins where the active power has an average standard deviation of 0.095/s, while the voltage deviates at 0.037/s. Lastly, Figure 9a(iv) illustrates ramp rate performance rendered at ZSI output where

the transient satisfies a requirement standard to prevent potential over-voltage due to inrush current; thus, ZSI only starts generating current after 15 s in a ramp-up fashion.

Thereon, further analyses were exhibited in Figure 9b, verifying ZSI's control algorithm which regulates boosting factor and DC–AC inverter processes. Figure 9b(i,ii) present the instantaneous AC(RMS) voltage generated at ZSI output. Figure 9b(iii) exhibits the boost voltage pulse signal generated by the Z-network and inverter's shorting proceedings. The magnitude profile of $Vdc_{boost}$ will fluctuate in real-time based due to inconsistent rectified voltage level generated by TST. Finally, Figure 9b(iv) refers to the voltage and current transient transpired at Z-network illustrating the controlled discharging process to render boosting sequence. A brief summary of ZSI performances was recorded in Table 4b at a selected tidal current speed.

**Table 4.** (**a**) ZSI $K_p$ and $K_i$ controller evaluations ($K_a$ = 0.05, $K_t$ = 0.1); (**b**) single ZSI operations with adaptive PI-controller ($K_a$ = 0.05, $K_t$ = 0.1) against arbitrary tidal speed profile (Std. Dev. 0.201). (**c**) comparisons on inverter performances in directing tidal energy conversion systems.

| (a) | | | | |
|---|---|---|---|---|
| **P-Gain** | **I-Gain** | **P–I Response** | **Settling** | **Standard** |
| $K_p$ | $K_i$ | | Time | Deviation |
| Low: 0.01–0.15 | Low: 0.01–0.15 | Oversht: 10.4% Overdamped | 16.3 s | 0.2728 Eff. 71.43% |
| Low: 0.01–0.15 | High: 1.50–3.89 | Oversht: 18.7% Diverged | nil | 3.4965 Eff. 11.47% |
| High: 1.50–3.89 | Low: 0.01–0.15 | Oversht: 29.1% Diverged | nil | 8.7423 Eff. 5.44% |
| High: 1.50–3.89 | High: 1.50–3.89 | Oversht: 1.85% underdamped | 9.56 s | 0.121 Eff. 89.45% |

| (b) | | | | |
|---|---|---|---|---|
| **Tidal Speed** | **3.50 kn** | **4.76 kn** | **5.89 kn** | **8.15 kn** |
| S.T. width: $T_Z$ $K_p$; $K_i$ | 93.666 µs 0.38–0.42; 1.2–2.1 | 55.666 µs 0.67–1.85; 1.98–2.76 | 49.333 µs 0.20–1.00; 2.02–2.31 | 49.233 µs 0.15–2.50; 1.5–3.0 |
| Z-network: $L_1 = L_2 = 0.0462$ H $C_1 = C_2 = 6.039$ µF $f_{ref} = 10$ kHz $f_{sw} = 30$ kHz | $Vdc_{IN}$: 24.06 kV $Vdc_{boost}$: 67.68 kV $Vac_{OUT}$: 50.116 kV $P_{out}$: 0.264 MW BF: 2.81 | $Vdc_{IN}$: 41.553 kV $Vdc_{boost}$: 69.49 kV $Vac_{OUT}$: 51.459 kV $P_{out}$: 1.283 MW BF: 1.67 | $Vdc_{IN}$: 48.218 kV $Vdc_{boost}$: 71.59 kV $Vac_{OUT}$: 53.013 kV $P_{out}$: 1.374 MW BF: 1.48 | $Vdc_{IN}$: 48.583 kV $Vdc_{boost}$: 71.80 kV $Vac_{OUT}$: 53.265 kV $P_{out}$: 1.380 MW BF: 1.47 |
| P–I Response | Settling: 15.7 s Overshoot (%): 0% Rise (10–90%): 10.45 s | Settling: 10.2 s Overshoot (%): 3.545% Rise (10–90%): 10.45 s | Settling: 10.6 s Overshoot (%): 7.067% Rise (10–90%): 10.45 s | Settling: 11.7 s Overshoot (%): 10.389% Rise (10–90%): 10.45 s |

| (c) | | | |
|---|---|---|---|
| **Control Method** | **Proposed** | **Marios [15]** | **Mahda [34]** |
| Switching Freq., (Hz) | 20 k | 7 k for optimum | 60 |
| Power ripple, (%) | >0.34%- <0.69% | >0.28%- <0.65% | <1% |
| Voltage THD, (%) | 1.89% | 2.53% | 3.72% |
| Controller Behaviour | Adaptive | Freq. Dependant | Fixed |
| PCC Busbar Freq. Deviation, (%) | 0.083% | Not mentioned | 0.072% |
| Fault-ride through | 120 ms | Not Tested | 230 ms |
| Converter Eff. (%) | 93.56- 98.73% | 89.67- 96% | 97.78- 99.3% |

Ultimately, engagement of OVC-Rect was disposed to perform active rectification before coupling to the bi-pole HVDC transmission. Figure 9c(i,ii) depicts the instantaneous voltage transient after being

filtered using a LCL-configured filter, designed to reduce high-frequency current harmonics absorbed by the diode-rectifier. Furthermore, the LCL filter generates higher resonance frequency attenuation and is thus suitable when engaging H-XFMR. The rectified DC voltage transient at the sending-end of HVDC transmission seen in Figure 9c(iii) shows a constant DC voltage level that allows maximum active power transference to receiving-side. The performance of Freq/Watt controller seen in Figure 9c(iv) experienced a frequency dip crisis during the transition of tidal direction, 25 Hz losses. Nevertheless, the controller was able to refrain from further dipping and regain stability instantaneously. In the case of Freq/Watt control not being activated, the system suffered continual frequency dipping.

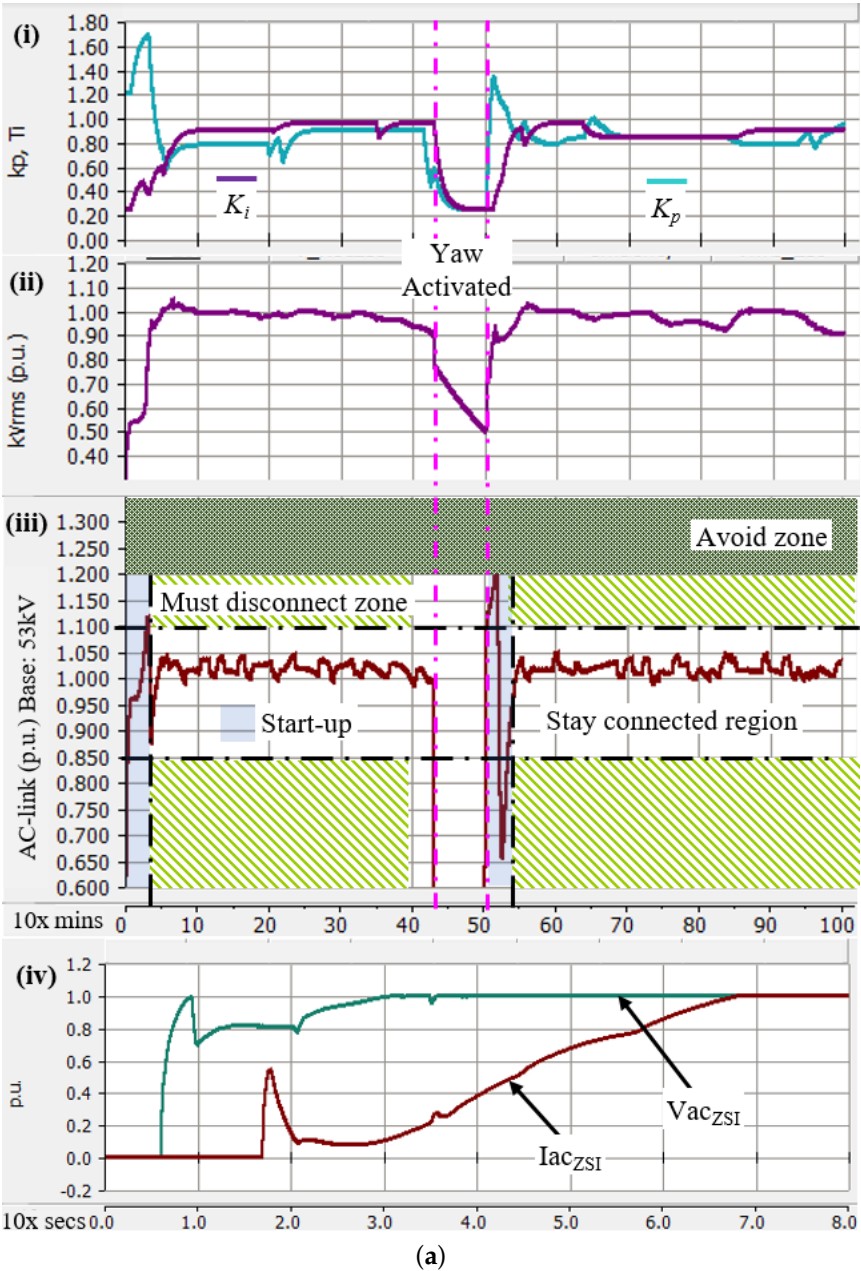

**Figure 9.** *Cont.*

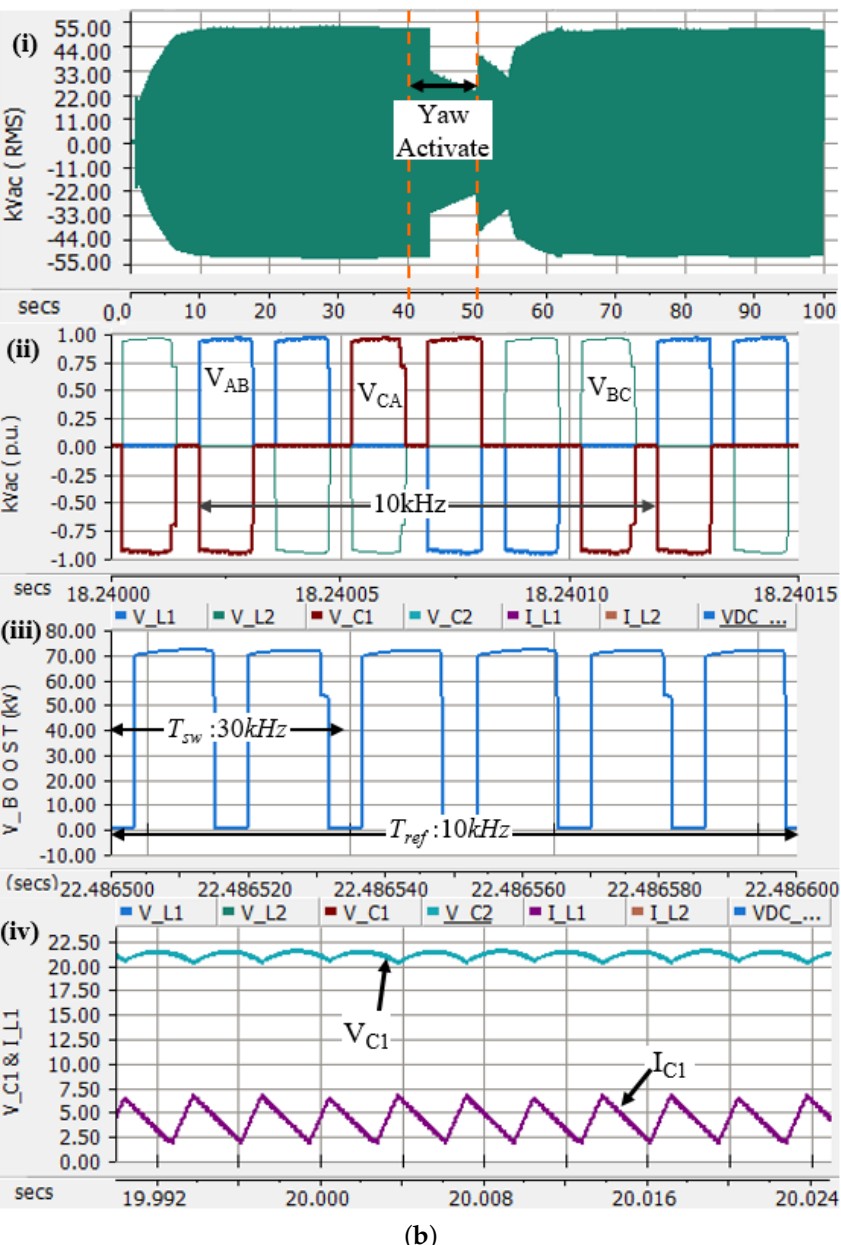

**Figure 9.** *Cont.*

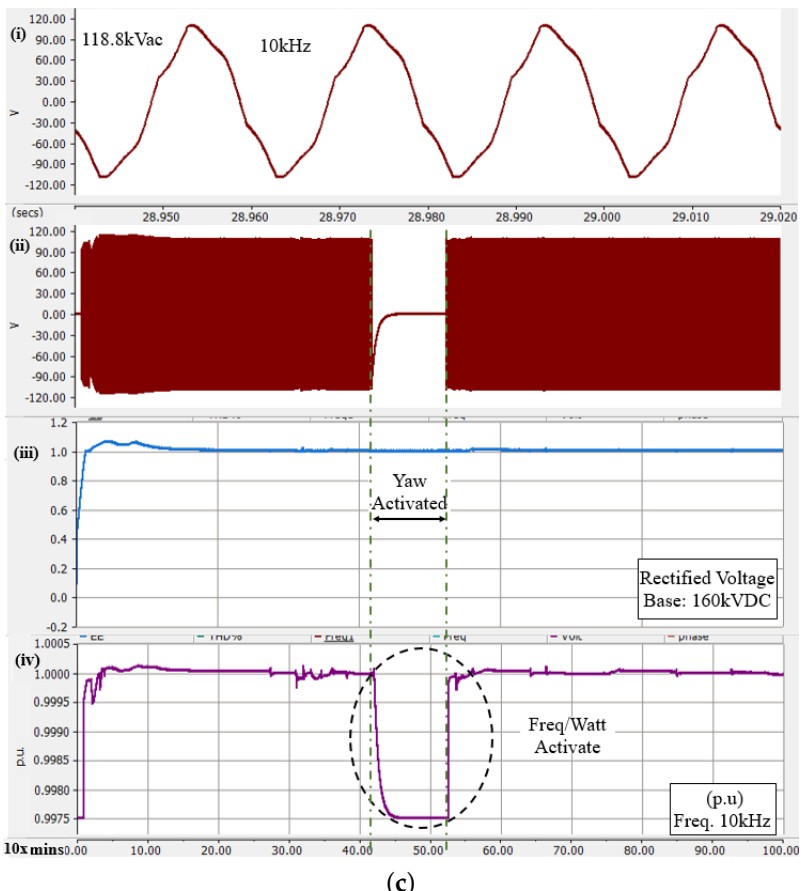

(c)

**Figure 9.** (**a**) pitching-PMSG-ZSI performances; (i) $K_p$ and $K_i$ adaptive responses; (ii) AC voltage transient generated from ZSI's output, ordered 53 kV *AC*; (iii) voltage transient comparison against voltage ride-through criterion, ZSI output; (iv) ramp rate of voltage and current transients during PMSG start-up; (**b**) ZSI operations; (i) instantaneous ZSI output voltage; (ii) zoomed in ZSI output voltage transient; (iii) boost voltage transient after Z-network; (iv) capacitor's voltage (kV) and inductor's current ($kA$), Z-network; (**c**) OVC-Rect operations; (i) instantaneous LCL-filtered AC voltage at secondary-side of H-XFMR; (ii) AC-link voltage (single-phase), $kV_{ab}(RMS)$; (iii) rectified DC voltage at HVDC transmission, p.u.; (iv) frequency response profile at secondary-side of H-XFMR.

To apprehend operational credibility of the proposed tidal energy conversion system for a single TST operation, analytical comparisons were presented in Table 4c against two other published methodologies [15,34]. In [15], C.S. Marios proposed a tidal current conversion system that uses a personalised filter design and controlled operating frequency for generator control. The paper's key focus was to proffer an alternate solution in integrating tidal turbines with the grid through variable-speed control strategies. Contrarily in [34], Mahda proposed a new converter control strategy for tidal turbine to address fast-changing tidal current speed profiles. Here, a back-to-back voltage source converter with a three-level neutral point coupling is used to govern the generator's variable-frequency drive responses.

### 5.3. DC Collection Point: Cascaded DCCAs Connecting to Bi-Pole HVDC Transmission

Corroborating results attained from operating a single TST system, subsequent investigations are focused into analysing performance serviceability of tying large-scale tidal stream farm with AC distribution (Figure 1). Here, chaotic turbulence regime and wave propagational delays are induced to

gain realistic dissipation of tidal current across a TST cluster—3-tidal channelling and induced mixed semidiurnal tided cycles on neighbouring turbines shown in Figure 10a.

Preliminary investigations focus on a single cluster deployment which is comprised of six aggregated TSTs coupled to a single DCCA, verifying transient responses at bi-pole HVDC-Link transmission. DC voltage level and power deliverance capacities were monitored to appraise the system's efficiency before cascading, with nine clusters remaining together. Table 5a depicts performance of a single TST cluster at separate tidal channels.

Figure 10b(i) demonstrates transient responses transpired at the DCCA AC-Link, 118 kV *AC*, where all six TSTs were coupled together. Regardless of disassociated operations at separate PMSG, dynamic synchronisation of voltage and frequency levels were rendered by respective ZSI. Such annexations will propagate a safe interoperability proceeding when cascading multiple TST clusters together and assuring a governed rectified DC voltage at the sending-end of the HVDC-Link. In Figure 10b(ii), two brief events of voltage dip lasted for 16 ms orderly are transpired due to generator start-ups TSTs at tidal channel 2 and 3. Thus, based on the voltage ride-through standards, the serviceability of TSTs was not interrupted or forced to be disengaged from the distribution network.

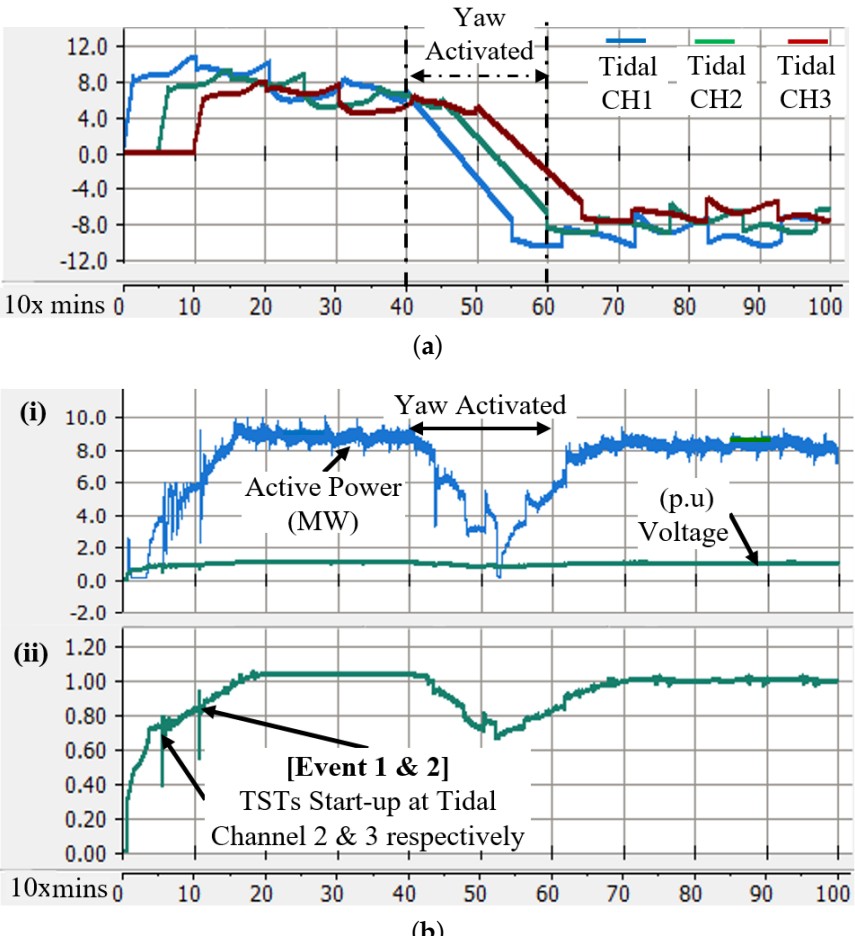

**Figure 10.** *Cont.*

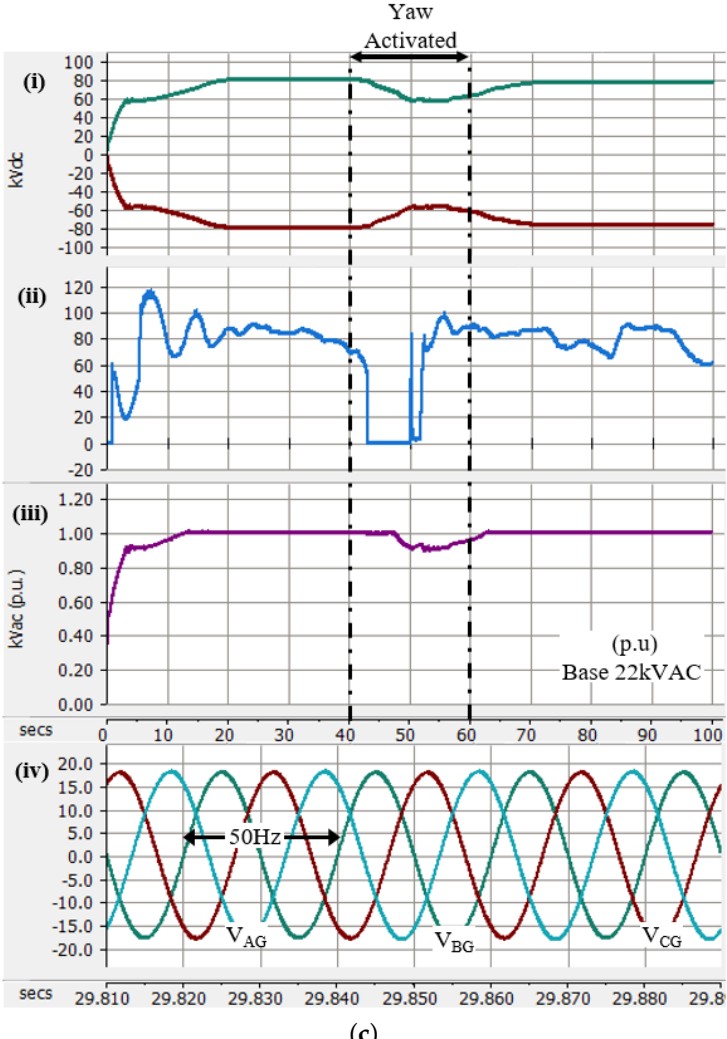

(c)

**Figure 10.** (**a**) tidal current profiles (*knots*) for three separate channels across six TSTs; (**b**) DCCA's AC-Link Busbar coupling six TSTs; (i) power (MW) and voltage profiles; (ii) voltage transient detecting two dipping events during TST start-ups; (**c**) transient responses at PCC Busbar and bi-pole HVDC transmission; (i) bi-pole HVDC-Link voltage (sending-end), $kVDC$; (ii) harvested active power yield from TST farm at PCC, MW; (iii) voltage transient at PCC busbar; (iv) instantaneous voltage transient at PCC busbar.

**Table 5.** (**a**) Performances of individual TST at respective tidal channels for cluster 1; (**b**) readings at PCC busbar and bi-pole transmission sending-end.

| (a) | | | | | |
|---|---|---|---|---|---|
| Tidal Channel 1 (Front Row) Turbulence Scale: 50 m Surface Drag Coeff.: 0.192 kn Gust Peak Velocity: 0.15 kn | | Tidal Channel 2 (Middle Row) Turbulence Scale: 100 m Surface Drag Coeff.: 0.130 kn Gust Peak Velocity: 0.09 kn | | Tidal Channel 3 (Last Row) Turbulence Scale: 150 Surface Drag Coeff.: 0.035 kn Gust Peak Velocity: 0.04 kn | |
| **TST 1** [Std Dev. 0.162] Tidal Noise 0.02 rad/s | **TST 2** [Std Dev. 0.187] Tidal Noise 0.16 rad/s | **TST 3** [Std Dev. 0.136] Tidal Noise 0.12 rad/s | **TST 4** [Std Dev. 0.175] Tidal Noise 0.24 rad/s | **TST 5** [Std Dev. 0.121] Tidal Noise 0.26 rad/s | **TST 6** [Std Dev. 0.187] Tidal Noise 0.3 rad/s |
| Generating Unit, PMSG | | | | | |
| $V_{PMSG}$ 0.938 p.u. $\omega_{PMSG}$ 11.925 rpm $I_{PMSG}$ 1.335 kArms $\tau_{PMSG}$ 0.996 p.u. | $V_{PMSG}$ 0.925 p.u. $\omega_{PMSG}$ 11.789 rpm $I_{PMSG}$ 1.313 kArms $\tau_{PMSG}$ 0.991 p.u. | $V_{PMSG}$ 0.803 p.u. $\omega_{PMSG}$ 10.245 rpm $I_{PMSG}$ 1.308 kArms $\tau_{PMSG}$ 0.865 p.u. | $V_{PMSG}$ 0.793 p.u. $\omega_{PMSG}$ 10.233 rpm $I_{PMSG}$ 1.311 kArms $\tau_{PMSG}$ 0.848 p.u. | $V_{PMSG}$ 0.634 p.u. $\omega_{PMSG}$ 8.950 rpm $I_{PMSG}$ 1.293 kArms $\tau_{PMSG}$ 0.778 p.u. | $V_{PMSG}$ 0.622 p.u. $\omega_{PMSG}$ 8.897 rpm $I_{PMSG}$ 1.290 kArms $\tau_{PMSG}$ 0.733 p.u. |
| ZSI | | | | | |
| $V_{IN}$ 45.05–45.75 kVDC $BF_{ave}$ 1.52 $T_z$ 50.66 µs $Vac_{OUT}$ 51.37 kVAC $THD_{AC\text{-}LINK}$ 1.246% $P_{out}$ 1.485–1.492 MW | $V_{IN}$ 43.03–43.87 kVDC $BF_{ave}$ 1.59 $T_z$ 53.00 µs $Vac_{OUT}$ 51.38 kVAC $THD_{AC\text{-}LINK}$ 1.123% $P_{out}$ 1.452–1.464 MW | $V_{IN}$ 37.95–38.65 kVDC $BF_{ave}$ 1.84 $T_z$ 61.33 µs $Vac_{OUT}$ 51.38 kVAC $THD_{AC\text{-}LINK}$ 1.346% $P_{out}$ 1.329–1.372 MW | $V_{IN}$ 35.88–36.41 kVDC $BF_{ave}$ 1.90 $T_z$ 63.33 µs $Vac_{OUT}$ 51.39 kVAC $THD_{AC\text{-}LINK}$ 1.245% $P_{out}$ 1.301–1.322 MW | $V_{IN}$ 31.64–32.42 kVDC $BF_{ave}$ 2.18 $T_z$ 72.66 µs $Vac_{OUT}$ 51.36 kVAC $THD_{AC\text{-}LINK}$ 1.269% $P_{out}$ 1.180–1.207 MW | $V_{IN}$ 31.48–32.16 kVDC $BF_{ave}$ 2.25 $T_z$ 75.00 µs $Vac_{OUT}$ 51.37 kVAC $THD_{AC\text{-}LINK}$ 1.279% $P_{out}$ 1.132–1.111 MW |

| (b) | | | | |
|---|---|---|---|---|
| **HVDC-Link (kVDC)** | **Voltage, (p.u.) (Base: 22 kVAC)** | **Active Power (MW)** | **T.H.D (%)** | **Frequency (Hz)** |
| ±79.92- ±80.13 | 0.992- 1.002 | 81.5- 87.88 | 0.121 | 49.96- 50.01 |

Decisively, Figure 10c and Table 5b exhibit results collected at the PCC busbar for 10 paralleled DCCAs coupled to the 22 kV *AC* distribution network. Figure 10c(i) illustrates the DC voltage transient at HVDC-link sending-end region while Figure 10c(iv) presents phase-frequency transients at PCC busbar governed by the grid-side VSC. As expected, promising results in synchronising voltage and frequency regulatory at PCC shown in Figure 10c(iii), with *THD* rated at less than 0.15% is attained. Each TST cluster has managed to generate a minimum active power rating of 8.4 MW seen in Figure 10c(ii) before transmitting over to the receiving-end.

*5.4. Voltage and Frequency Ride-through Performances during Momentary Fault Interruption*

In this case study, random TSTs are deliberately routed to offline, simulating permanent faulted TSTs or scheduled to shut down for maintenance. Here, transient analyses and grid-tied regulatory are monitored to ensure safe engagements between remaining online TSTs and the distribution network (grid). Testing was done in a controlled environment where the tidal current velocity is constant at 5.67 kn to view distinctive behavioural. The investigation is broken down into twofold; (i) tripping of two TSTs in a cluster across all 10 DCCAs, and (ii) tripping of all six TSTs only in a single cluster (i.e., cluster No. 5).

5.4.1. Tripping Two TSTs in a Cluster for All 10 DCCAs

Analyses were focused on the voltage and frequency transients at respective DCCA AC-Link busbar. It aims to appraise efficiencies and power quality of proposed control schemes in DCCA for the remaining online TSTs' tidal energy conversion systems. Figure 11a reflects the voltage transients at a single DCCA AC-link busbar at cluster-1. A momentary voltage dip for 50 $\mu$s was noticed where it falls below 80%. Nevertheless, with the Volt/Watt controller in DCCA and ZSI's shoot-through control, the voltage and frequency ride-through function, respectively, were activated, allowing remaining TSTs in the cluster to stay connected considering voltage and frequency transients at HVDC transmission and PCC busbar met the threshold limit requirements suggested in IEEE 1547-2018, Figure 11b. Table 6 summarizes time taken for the system to regain equilibrium and impacts on voltage and frequency transient, and power qualities.

**Table 6.** Post-measurements collected at TST cluster-1 and PCC busbar (20 TST Units Offline).

| Sending-End Region | | | |
|---|---|---|---|
| Voltage TST AC-Link 0.75–1.01 p.u. | Frequency TST AC-Link 19.89–20.2 kHz | Time taken system recover 0.62 ms | Voltage dip sag by 15% for 0.5 cycle |
| **Receiving-End Region** | | | |
| HVDC-Link 158.8–160 $kVDC$ | Voltage PCC Busbar 0.99–1.01 p.u. | T.H.D PCC Busbar 1.28–1.535% | Active Power $P_{ONSHORE}$ 54.67 MW |

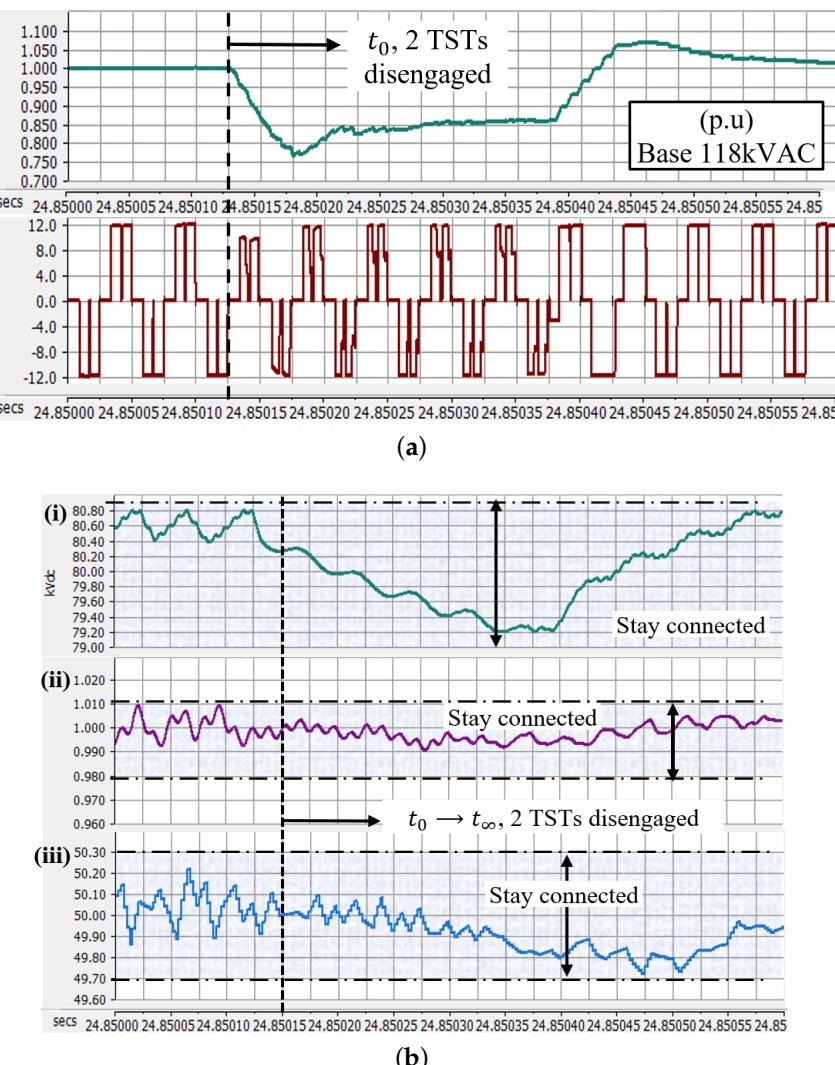

**Figure 11.** (**a**) measured RMS voltage and instantaneous voltage at DCCA AC-Link busbar upon disconnecting 2 TSTs in a cluster; (**b**) transient signals at bi-pole HVDC and PCC busbar upon disconnecting TSTs; (i) bi-pole (+DC) voltage profile at sending-end region, ($kV\ DC$); (ii) voltage transient at PCC busbar, (p.u.); (iii) frequency transient at PCC busbar, (Hz).

### 5.4.2. Tripping a Single TST Cluster from the DC Collection System

Conversely, this case study dealt with disengaging a single DCCA from HVDC transmission. Analyses were drawn towards the DC voltage level at the $\pm 80\ kVDC$ bi-pole HVDC transmission while ensuring that voltage and frequency synchronicity at the PCC busbar are annexed.

Figure 12 expressed commendable voltage and frequency responses, in compliance with the grid-tied IEEE 1547-2018 standards. The system neither suffered voltage sag crisis at HVDC transmission nor large frequency deviations at the PCC busbar, refraining from malicious triggering of a circuit breaker at the PCC busbar. Overall, the tidal stream farm took approximately 400 ms to regain stability and discharge voltage $THD$ percentage of 2.78% at the PCC busbar after disengaging TST cluster-5.

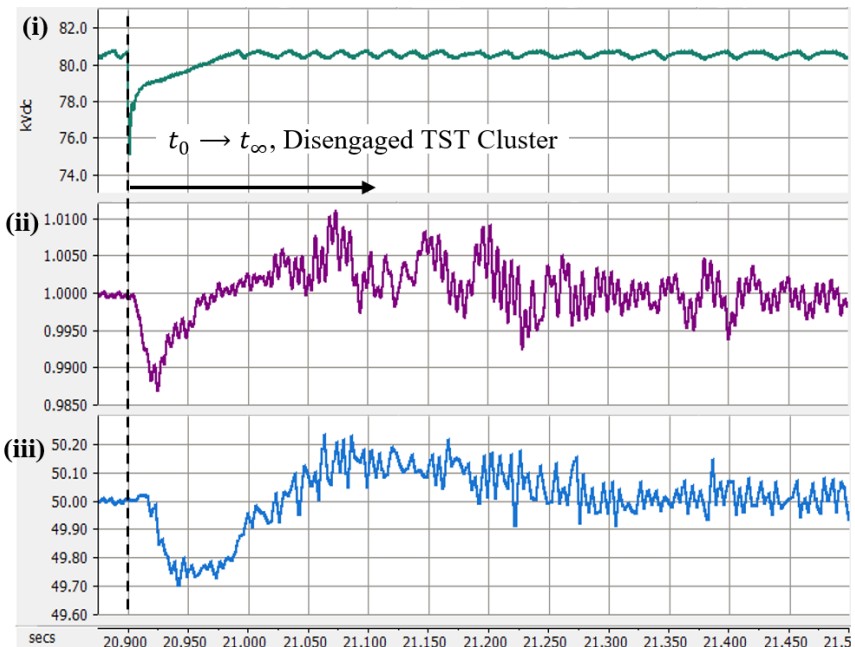

**Figure 12.** Bi-pole HVDC transmission and PCC busbar transients upon tripping TST cluster-5. (i) bi-pole (+DC) voltage at sending-end region, $(kVDC)$; (ii) voltage transient at the PCC busbar, (p.u.); (iii) frequency level at the PCC busbar, (Hz).

### 5.5. Summary of Proposed System Performances against Other DC Collection System Methodologies

With limited research analyses on large-scale deployment for TST engagements, alternatively, comparisons were made against HVDC-based offshore wind farm technology. They have showed relevancies in designing control strategies for energy conversion systems, modelling of offshore DC collection point, ride-through function compliances, and direct drive turbine physique operations when tied to the distribution network. Table 7 presents three prepositions to view respective system's superiority and operation stability against the proposed tidal stream farm. Evaluations focus on control architectonic, electrical design configuration, and ride-through capability. In [29], Liang proposes a study on operation and control of multiterminal HVDC transmission for offshore wind farm. Subsequently in [30], Raza presented a paper on coordinating a VSC-based multiterminal HVDC transmission system and lastly, in [35], Shi suggested an improved variable speed control for a series-connected collection system connecting to an HVDC transmission system.

The results justify that the proposed energy conversion and DC collection systems are adequate at preserving operational serviceability and administrations. Extending its active-rectification technologies to curtail manufacturing costs and alleviate control complexity, the proposed methodology contributes a possible revolutionary in governing the future's large-scale offshore AC-based renewable system.

**Table 7.** Performance comparisons against other control schemes and collection system design for large-scale deployment—ECS: Energy Conversion System.

| Methods | Proposed | Liang [29] | Raza [30] | Shi [35] |
|---|---|---|---|---|
| | | Generator-side Transmission | | |
| ECS Efficiency, (%) | 92.23–98.34 | 89.44–97.67 | 94.81–98.94 | 85.46–94.53 |
| Voltage Ripple, (%) | 1.77–1.98 | 1.34–1.79 | 1.67–1.92 | 1.77–2.15 |
| Frequency Std. Deviation | 0.116–0.124 | 0.149–0.162 | 0.167–0.173 | 0.136–0.152 |
| Generator Start-up | 8.18 s transient to steady-state | 6.8 s transient to steady-state | Not Mentioned | 14 s transient to steady-state |
| | | Receiving-side Transmission (PCC Busbar) | | |
| Frequency Std. Deviation | 0.12–0.125 | 0.116–0.121 | 0.127–0.134 | 0.121–0.142 |
| Voltage signal-to-noise ratio, $SNR$ | 97.67 dB | 98.12 dB | 97.81 dB | 97.13 dB |
| Power losses, $\approx$(%) | DC Line 200 km 10.3–14.12 | DC Line 100 km 7–11 | DC Line 100 km 15 | Disregard Transm Line Analyses |
| Power T.H.D, (%) | 0.23 | 0.143 | 0.321 | 0.276 |
| Fault-ride through abled | YES | YES | YES | Not Analysed |
| Voltage dip at Turbine, (%) | 10.43–12.45 | Not Analysed | 20.8–24.6 | Not Analysed |
| Time taken system recovery, (ms) | 50–70 | Not Analysed | 70–100 | Not Analysed |
| ECS Converter | Rec-ZSI | VSC | VSC | VSC |
| Controller Approach | PI Adaptive Controller | PI Droop Controller | P Droop Controller | P–I Controller |

## 6. Conclusions

This paper has presented analytical evaluations on both technical modelling and financial viewpoints when operating large-scale offshore tidal stream turbine farm in an HVDC-configured transmission network. The proposed DC Collection system is designed with a multimodular tidal energy conversion system, DCCA that services clustered direct drive TSTs. It aims to minimise rectification processes when compared to other suggested DC collection point that uses BtB-VSC topology. In addition, the proposed control systems in DCCA employs Volt/Watt and Watt/Speed functions to procure MPPT operation for TST and observes fault ride-through requirements mentioned in IEEE 1547-2018. The DCCA was modelled based on three-stage high frequency DC–DC conversion (SST), employing passive rectifier, ZSI with H-XFMR, and OVC rectifier (in chronological order). The ZSI control system is designed to operate in isochronous mode where Volt/Watt function is incorporated using an adaptive PI-controller. Correspondingly, as Volt/VAR controller was omitted, torque-controlled blade pitch with Watt/Speed function was proposed to create reserve active power generation to compensate over- or under-voltage crises, which results in close to MPPT operations. The approach ensures direct sole governance for PMSGs' current–speed–torque relationships in parallel with ride-through capabilities. The results have shown positive voltage and frequency transient response under various test cases: (i) controlled and arbitrary tidal current velocity profiles, (ii) three tidal channels incorporating wake, turbulence, and migration delay for array formatted deployment of TSTs, and (iii) tripping of two TSTs in a cluster and a cluster to stimulate momentary fault interruption or switched offline for maintenance. Moreover, the analyses are involved mainly with the tidal energy conversion performances ensuring that maximum power deliverance was attained from offshore to onshore, and compliance of voltage and frequency transients generating within the threshold limits for ride-through requirements.

Succeedingly, further evaluations on comprehending optimal sizing of each TST cluster and equating installation cost benefits based on a truncated DC collection system were correlated to adhere superior levelised energy costs against another established clustering topology. It is projected that the proposed TST farm can break even within 20 years by selling its energy for less than four cents per kilowatts. The results have shown that the number of TSTs in a cluster can influence turbines' MPPT efficiencies and DCCA's control structure (PI-controller) when handling larger TST cluster size. Hence, the trail and error approach was inferred where the results showed that six aggregated TSTs in a cluster gained superior engagements—installation costs and MPPT efficiency intersects.

Conclusively, the proposed DC collection system has shown potential eco-technological solutions in deploying the future's offshore large-scale AC-based power generation resources in the HVDC transmission system. The proposed tidal stream farm was designed based on industrial-ready specifications undergoing listed conscientious case studies to evaluate its deployment practicability. The modelled control features have displayed full governance in directing a safe and reliable grid connected operations at PCC and TST under the constraint of IEEE 1547-2018 Standard regulations. For future control implementations, the proposed DCCA can serve as a dynamic harmonic cancellation. It can autonomously refine settings to gain optimal voltage total harmonist at its terminal during steady state operations, imparting dynamic voltage support even when generation sources have no input power.

**Author Contributions:** Conceptualisation, M.R.B.M.S.; methodology, M.R.B.M.S.; software, M.R.B.M.S.; validation, M.R.B.M.S. and W.P.Q.T.; formal analysis, M.R.B.M.S.; investigation, M.R.B.M.S.; resources, T.N.R.; data curation, M.R.B.M.S.; writing—original draft preparation, M.R.B.M.S. and W.P.Q.T.; writing—review and editing, M.R.B.M.S., W.P.Q.T., and T.N.R.; visualisation, M.R.B.M.S. and W.P.Q.T.; supervision, T.N.R.; project administration, M.R.B.M.S. All authors have read and agreed to the published version of the manuscript.

**Funding:** This research received no external funding.

**Conflicts of Interest:** The authors declare no conflict of interest.

## Abbreviations

The following abbreviations are used in this manuscript:

| | |
|---|---|
| TST | Tidal Stream Turbine |
| VSC | Voltage Source Converter |
| HVDC | High-Voltage Direct Current |
| DCCA | DC Collection Aggregator |
| ZSI | Impedance Source Inverter |
| OVC-Rect | Over-Voltage Control Diode Rectifier |
| PMSG | Permanent Magnet Synchronous Generator |
| Watt/Speed | Active Power-Mechanical Speed |
| Volt/Watt | Voltage-Active power |
| Freq/Watt | Frequency-Active power |
| H-XFMR | High Frequency Transformer |
| PWM | Pulse Width Modulation |
| MPPT | Maximum Power Point of Tracking |

## Appendix A. Proof of Lemma for Basic Operations of ZSI

Strategic calculations are needed when estimating values of the reactive components in ZSI. As proposed, defining the minimum input voltage of the converter must be pre-identified as it provides an indicator where the boost factor and the current stresses of the components become maximal. Thus, the inductor's current operating region can be calculated using Label (A3). While Label (A1) and Label (A2) depict the formula in calculating the inductor and capacitor ratings based on current limit and ripple, respectively:

$$L_1 = L_2 = \frac{T_z \times (Vdc_{WT} + Vdc_{boost})}{2 \times (I_{L_{max}} - I_{L_{min}})} \tag{A1}$$

$$C_1 = C_2 = \frac{I_{input} \times T_z}{\left(\frac{Vdc_{WT} + Vdc_{boost}}{2}\right) \times ripple\%} \tag{A2}$$

$$\begin{aligned} I_{L_{max}} &= I_{input} + (I_{input} \times ripple\%) \\ I_{L_{min}} &= I_{input} - (I_{input} \times ripple\%) \end{aligned} \tag{A3}$$

where $I_{L_{max}}$ and $I_{L_{min}}$ dictate the maximum allowable peak-to-peak ripple current through the Z-inductor during shoot-through episodes. $T_z$ refers to the shoot-through period where instances of voltage surges generated from the DC–AC inverter will be stored into the Z-capacitor, thus boosting the Z-network's output DC voltage.

The shoot-through mode transacts two-state gating sequences when composing the 6-pulses' PWM signals: shoot-through state where both switching devices in the same phase are gated concurrently to create a short-circuit network. Secondly, the non shoot-through state operates similar to a typical Sinusoidal-PWM is employed. Figure A1 represents the control scheme in infusing switching pulses with a shoot-through component where $T_{ref}$, $T_{sw}$, $T_z$ denote the system operating, gating switching, and shoot-through period, respectively. $V_{ab}$, $V_{bc}$, and $V_{ca}$ are line-to-line voltage while $Vdc_{boost}$ measures the boost voltage at Z-network output. To pilot the shoot-through state sequences, the two parallel lines indicating the envelop boundaries were introduced—$Vupp_{ref}$ and $Vbott_{ref}$. The shoot-through period will be added together with the signal generator labelled *switching period w/shoot–through cycle*:

$$T_{sw} = \frac{1}{f_{sw}} = 3 \times T_{ref} \tag{A4}$$

$$BoostFactor(B.F.) = \frac{Vdc_{boost}}{Vdc_{WT}} \tag{A5}$$

$$Vupp_{ref} = 1 - \frac{B.F. - 1}{2B.F.}$$
$$Vbott_{ref} = -1 + \frac{B.F. - 1}{2B.F.} \tag{A6}$$

$$ST_{state} = \begin{cases} 1 & V_{sw} > Vupp_{ref} \\ 1 & V_{sw} < Vbott_{ref} \\ 1 & V_{ref} = 1 \\ 0 & \text{otherwise} \end{cases} \tag{A7}$$

$$T_z = T_{sw} \times \frac{B.F. - 1}{2B.F}, \tag{A8}$$

where $T_{sw}$ is the sawtooth-carrier signal period that yields a linear voltage conversion process. Continual inequality comparisons between $Vupp_{ref}$, $Vbott_{ref}$ and $V_{sw}$ are assessed to originate shoot-through sequences, $ST_{state}$, which causes both switching gates in the same phase to be turned on concurrently. The adopted system specifications are presented in Table A1.

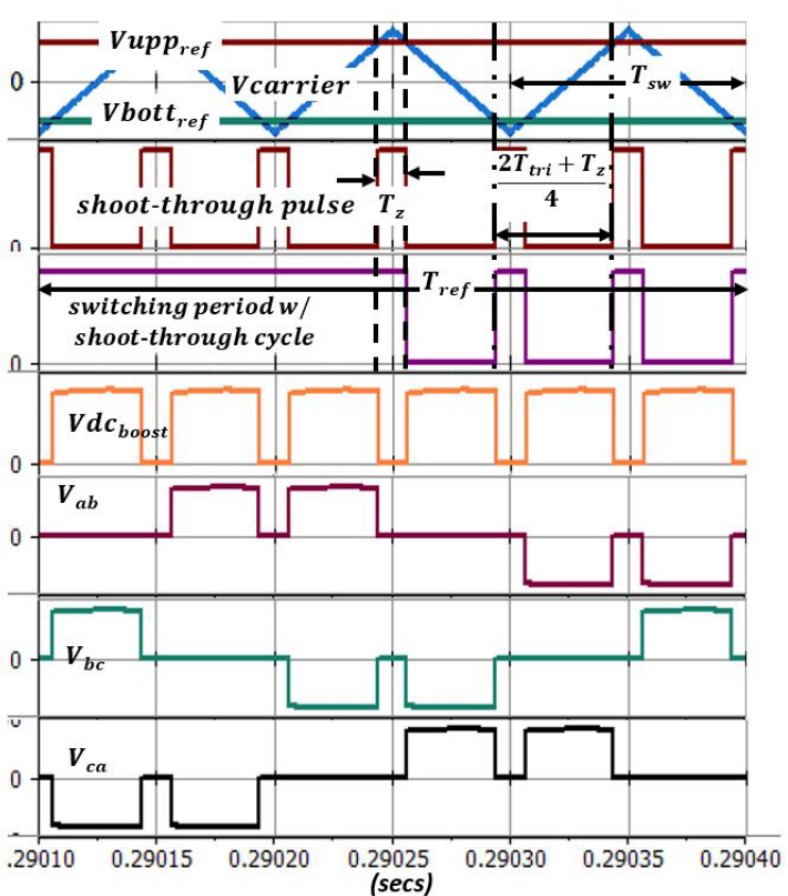

**Figure A1.** Switching PWM sequences for ZSI operation.

**Table A1.** Specifications and ratings of proposed devices.

| Permanent-Magnet Synchronous Generator Specifications | |
| --- | --- |
| Generator type | Radial Flux |
| Rated MVA | 1.5 *MVA* |
| Rated Voltage $(L - L)$ | 4.16 *kVAC* |
| Rated Frequency | 30 *Hz* |
| No. of pole pairs | 145 |
| Angular Moment of Inertia $(J = 2H)$ | 1.5 s |
| Generator type | Radial Flux |
| Tidal Steam Turbine Physique Specifications | |
| Turbine type (3-blades) | Horizontal-axis |
| Machine nominal angular mech. speed | 1.466 *rad/s* |
| Rated turbine power | 1.5 *MW* |
| Blade Radius | 9 *m* |
| Turbine Height | 12 *m* |
| MPPT tidal current velocity region | 5.956–11.07 *kn* |
| Pitch Elevation | 0–30 *deg* |
| DED Converter Parameters, Smart Aggregator Inverter (SAI) | |
| Six-pulse Frequency Control Diode Rectifier | |
| Forward voltage drop, $V_{DROP}$ | 4.49–5.35 *V* |
| Maximum average output current, $I_{MAX}$ | 56.25$A_{DC}$ |
| Switching Freq. Chopper | 1650 *Hz* |
| Impedance Source Inverter | |
| Output power, $P_{OUT}$ | 1.5 *kW* |
| Input Voltage, $V_{IN}$ | 23.33–70 *kVDC* |
| Constant Boost Voltage, $V_{BOOST}$ | 70 *kVDC* |
| Output voltage, $V_{OUT}$ | 51.41 *kVAC* |
| Output frequency | 10 *kHz* |
| Current ripple z-network inductors | 60% |
| Voltage ripple z-network capacitors | 3% |

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
