# Peer review of "Design and Control of a DC Collection System for Modular-Based Direct Electromechanical Drive Turbines in High Voltage Direct Current Transmission"

_electronics, doi:10.3390/electronics9030493_

Round 1
Reviewer 1 Report
The text of this paper is very difficult to understand. English needs very serious improvement. We don't know what "... riveting 15% of ..." (line 25) means? We don't know what "... ocean harvesting methods ..." (line 27) means?
Line 102 should say: „… at 80 kV DC …”. Line 144 should say: „… mutual inductances …”. Line 145 should say: „… the stator inductance …”. Line 145: magnetic flux in Wb (not Wb/m2). An inductor it is a coil used to introduce inductance into an electric circuit. The inductance it is the property of an electric circuit by which an electromotive force is induced in it as the result of a changing magnetic flux.
“PMSG was elected as …” (first line in Section 2.3)? “; proposed DED”?
Formulas (2) - (7) are very well known. No relation between the electromagnetic and the electromechanical section of TST system.
Author Response
Dear Editor
Please refer to the attached documents: Response to Reviewer 1.
Thank you.
Kind Regards
Muhammad Ramadan and et al.

Reviewer 2 Report
„Design and Control of DC Collection System for Modular-based Direct Electromechanical Drive Turbines in High Voltage Direct Current Transmission” for the Journal Electronics.
The paper „Design and Control of DC Collection System for Modular-based Direct Electromechanical Drive Turbines in High Voltage Direct Current Transmission” intends to promotes an initiative for offshore DC collection system that accommodates advancement in high voltage ride-through features to address high penetration of tidal stream turbines in isochronous operation.
The paper consists of six sections: Introduction, Proposed Testbed System: 90MW Tidal Stream Farm Configured in HVDC Transmission Network, Proposed Control Algorithms, Cluster Sizing & Capital Investments, Simulation Results and Conclusions.
The abstract must contains the main purpose of the paper, the research method used in the research and the main contributions.
It would be very useful to add in the "Introduction" section the purpose, objectives and hypothesis of the research. We consider that the introduction should specify the novelty of the paper compared to other papers published in this area. The authors should refer to other papers published in the Electronics Journal. Also, we consider the literature is not enough and that is why we recommend the authors to refer to other recent works indexed in Web of Science, Scopus, Emerald, Cambrige etc. We consider the literature review and introduction should be presented as separate sections. The conclusions must to be developed, because a single conclusion is not relevant to this research.
Author Response
Dear Editor
Please refer to the attached document: Response to Reviewer.
Thank you.
Kind Regards
Muhammad Ramadan and et al.
